



# Understanding drivers and biases of simulated CO emissions by the INFERNO fire model over South America

Maria P. Veláquez-García[1,2], Richard J. Pope[1,2], Steven T. Turnock[3,4], Chetan Deva[1], David P. Moore[5,6], Guilherme Mataveli[7,8], Steve R. Arnold[1], Ruth M. Doherty[9], and Martyn P. Chiperffield[1,2]

[1]School of Earth and Environment, University of Leeds, Leeds, UK
[2]National Centre for Earth Observation, University of Leeds, Leeds, UK
[3]Met Office Hadley Centre, Exeter, UK
[4]Met Office@Leeds, University of Leeds, UK
[5]Department of Physics and Astronomy, University of Leicester, Leicester, UK
[6]National Centre for Earth Observation, University of Leicester, Leicester, UK
[7]Earth Observation and Geoinformatics Division, National Institute for Space Research, São José dos Campos, Brazil
[8]Tyndall Centre for Climate Change Research, School of Environmental Sciences, University of East Anglia, Norwich, UK
[9]School of GeoSciences, University of Edinburgh, Edinburgh, UK

**Correspondence:** Maria P. Veláquez-García (eempvg@leeds.ac.uk)

**Abstract.**

Integrating fire simulation into climate models enhances our understanding of ecosystem-fire-climate interactions, clarifying the role of fire in the carbon cycle and other processes. The Interactive Fires and Emissions algorithm for Natural Environments (INFERNO) is one of the new modules in the upgraded UK Earth System Model (UKESM). Here, we use a version of INFERNO coupled only with the Joint UK Land Environment Simulator (JULES) to evaluate its performance and biases over South America (SA); a region that accounts for ∼15% of global fire carbon emissions. For this, we compared carbon monoxide (CO) estimates from INFERNO (2004-2021) with five satellite-based biomass-burning inventories, conducted sensitivity experiments and developed a machine learning (ML) model targeting biases. INFERNO was able to represent CO emissions in most of the fire-active zone in SA, particularly the southern Amazon 'Arc of Deforestation', but overestimates emissions (∼100%) outside them (e.g. within the Amazon forest). The ML model ($R^2 = 64\%$) indicates that tree categories of Plant Functional Types (PFTs) and soil moisture— through its role in flammability and gross primary productivity (GPP) —significantly influence spatiotemporal biases. In northern SA, CO emissions were overestimated by approximately 300% due to seasonal cycle inaccuracies, while INFERNO showed lower biases in southern SA emissions despite lacking seasonal representation. Both flammability and GPP underpinned the limited simulation of the seasonal cycle. Although INFERNO misrepresented emissions trends in the Arc of Deforestation, it successfully captured the increase in emissions in the eastern Andean Mountains from 2014 to 2021, albeit underestimating their magnitude. Sensitivity experiments revealed that the underlying PFT affected spatiotemporal variability (115%) and trends (167%) in CO emissions, while flammability influenced the seasonal cycle (116%) and trends (158%). These findings highlight the need for enhanced PFT accuracy and a deeper understanding of the roles of precipitation/soil moisture in GPP and flammability, as well as the consideration of landscape fragmentation to represent land management and forest fire vulnerability.



## 1 Introduction

Wildfires (both natural and anthropogenic) are becoming increasingly frequent and intense in key ecosystems around the world, largely due to climate change and land use practices (Cunningham et al., 2024; Zheng et al., 2021). The recent increase in wild-
fires in tropical and boreal forests has led to a rise in carbon emissions from fires, despite the ongoing decline in global burned area (Zheng et al., 2023, 2021). Due to the lack of resilience of these ecosystems to fires, the success of fire-prone ecosystems is enhanced over the burned area. As a result, these fires cause forests to shift to a negative carbon balance, i.e. becoming a net source of carbon (Yue et al., 2016). Annually, fires emit 7.3 Pg of $CO_2$ to the atmosphere (van der Werf et al., 2017), along with large quantities of other greenhouse gases such as methane ($CH_4$) and nitrous oxide ($N_2O$) (Heilman et al., 2014). The
effects of fire emissions, however, become complex with the additional release of large amounts of aerosols, including black carbon and organic aerosols, which have different interactions with clouds, radiation, precipitation and atmospheric circulation (Magahey and Kooperman, 2023; Thornhill et al., 2018; Wu et al., 2011). Other air pollutants, which can be precursors to ozone ($O_3$) formation, such as nitrogen dioxide ($NO_2$), are also emitted from fires. With this, fire emissions can adversely impact ecosystems through $O_3$ stress, which affects plant growth (Pacifico et al., 2015). Additionally, fire emissions can alter
the biogeochemical cycles of certain key elements for plant growth, such as carbon, phosphorus, nitrogen, and iron (Bauters et al., 2018; Hamilton et al., 2022). Humans significantly influence and are influenced by ecosystem-fire-climate interactions. These interactions can affect ecosystem functions, crop yields, and overall human health. The increase in air pollutants from wildfires has led to higher hospitalisation rates, particularly among children (Arrizaga et al., 2023), as well as human lung cell damage (de Oliveira Alves et al., 2017), and cases of low birth weight (Candido da Silva et al., 2014).

The limited representation of this complex climate-fire-ecosystem interaction in Earth System Models (ESMs) has led to large uncertainties in future climate projections (Canadell et al., 2021; Kloster and Lasslop, 2017; Hanan et al., 2022). This issue is further exacerbated by ESMs often lacking an interactive fire model component to represent the coupling of fire, land, atmosphere and climate interactions (Lasslop et al., 2019). There is particularly high uncertainty of the impact of fire on biogeochemical cycles and, in turn, the contribution to ecosystem change and climate (Lasslop et al., 2019). Multiple
coupled models have attempted to understand the complex climate-fire-ecosystem interactions in the last few decades (Hantson et al., 2016). Still, including fire models within global climate models is a growing and challenging development. In fact, the number of fire-related variables submitted by ESMs in the sixth Coupled Model Intercomparison Project (CMIP6) has doubled compared to its predecessor, CMIP5 (Hantson et al., 2016; Li et al., 2024). Additionally, efforts to enhance our understanding of fire processes and their representation in global models are being enhanced through collaborations, such as the Fire Modelling
Intercomparison Project (FireMIP) (Li et al., 2019; Hantson et al., 2020).

To improve the representation of climate-fire-ecosystem dynamics within CMIP7 experiments, the United Kingdom Earth System Model (UKESM) will couple the Interactive Fires and Emissions algorithm for Natural Environments (INFERNO) (Mangeon et al., 2016) to its vegetation model (the Joint UK Land Environment Simulator - JULES) and atmospheric model





(including coupling with the UK Chemistry and Aerosols Model - UKCA). INFERNO coupled to JULES has already partici-
pated in FireMIP, where INFERNO provided an accurate representation of global burned area and carbon emissions (Hantson
et al., 2020; Teixeira et al., 2021) and outperformed most of the studied coupled models in simulating the spatial patterns of fire
carbon emissions (Hantson et al., 2020). However, the model has also faced challenges in different parts of the world, such as
South America (SA), where INFERNO has been limited by the complexity of socioeconomic and political influences on land
management within the region (Burton et al., 2022). In general, FireMIP showed differences between models and a frequent
overestimation of emissions in this region, where anthropogenic influences on fires (e.g., deforestation) are the main challenge
(Li et al., 2019; Hantson et al., 2020). Enhancing global model performance in SA is crucial, as the region accounts for 15%
of annual fire carbon emissions. Furthermore, changes in the carbon balance and land cover in the Amazon can provoke sig-
nificant regional and global effects (Snyder, 2010; Zhou et al., 2021; Wang et al., 2023). Moreover, parts of the Amazon have
already become a carbon source due to deforestation and climate change (Gatti et al., 2021).

The current research on simulated fire emissions from INFERNO over SA lacks a thorough evaluation of the sensitivity and
biases of the model. This study decisively addresses this gap by rigorously assessing the performance of the model, as well as
examining the sensitivity of simulated fire emissions and the biases associated with various model processes and parameters.
For this, we compare the carbon monoxide (CO) emissions from fires simulated by JULES-INFERNO with various biomass
burning inventories and satellite-retrieved total column CO (TCCO). We focus on CO due to its significant emission rate from
fires and the availability of complementary satellite missions that retrieve atmospheric TCCO. In SA, TCCO has been suggested
as a valuable addition to fire activity monitoring since fire is the main source of CO in the region (Naus et al., 2022; Jury and
Pabón, 2021). Additionally, inventories generally align well with CO emissions in SA (Liu et al., 2020; Hua et al., 2024). To
describe the sensitivity of the estimated CO emissions and their biases, we used sensitivity experiments and a machine learning
(ML) approach.

## 2 Data and methods

This study utilised five biomass burning inventories to evaluate INFERNO CO emissions simulations in terms of spatiotemporal
distribution, seasonal cycle, and regional trend in SA from 2004 to 2021. To support the assessment, we also used TCCO
retrievals. The inventories and TCCO products are introduced in Section 2.2 and 2.3, respectively. In Section 2.4, we provide
a brief overview of the JULES-ES setup of the third simulation round of the Inter-Sectoral Impact Model Intercomparison
Project (ISIMIP3). This section also presents the JULES-INFERNO coupling and the key equations that it uses. Throughout
the study, the JULES-ES model using the ERA5 reanalysis served as the control model for most of the comparisons with
inventories. However, we conducted multiple sensitivity experiments, presented in Section 2.5, that modify the representation
of different processes within JULES-INFERNO. These experiments were compared to the inventories and the control run. We
also developed a machine learning (ML) model to explain CO emissions biases in SA in order to analyse the processes driving
the INFERNO biases.



## 2.1 Study area

This study focuses on continental SA and uses a regional classification that divides SA into three areas: northern SA (North-SA), central SA (Mid-SA) and southern SA (South-SA). North-SA consider the territory from latitude 0.5°N to the continental boundary between Colombia and Panama, Mid-SA ranges from 17.5°S to 0°, and South-SA ranges from 55°S to 17°S ( see Fig. S1). This classification aligns with the method used by Li et al. (2024), which identifies the "Arc of Deforestation" zone, here referred to as Mid-SA, separating it from the remaining southern hemisphere SA. The region commonly defined as the "Southern Hemisphere South America - SHSA" (van der Werf et al., 2017) encompasses both Mid-SA and South-SA. To facilitate a comparative analysis, we will refer to it as Mid-SA/South-SA when convenient.

Multiple factors are responsible for wildfires in SA, both of natural and anthropogenic origin. Yet, most are directly related to anthropogenic activities in highly fire-prone ecosystems, such as Cerrado and Llanos, and sensitive ecosystems, such as the Amazon rainforest (Chen et al., 2013; Menezes et al., 2022; van der Werf et al., 2010). In SA, frequent fire occurrences are mainly concentrated in the transition forest region, recognised as the "Arc of Deforestation." This area is not only highlighted by the continued land-use conversion but also for being the world's largest savanna-forest transition (Marques et al., 2020). The Arc of Deforestation in this study is located in Mid-SA, and includes the deforestation front of Bolivia, Brazil and Perú. This study assessed CO emissions on deforestation fronts and particular ecoregions of SA using the shapefiles provided in Pacheco et al. (2021) and Dinerstein et al. (2017), respectively. These zones are displayed by Fig. S1b.

## 2.2 Biomass burning emission inventories

To evaluate the simulated CO, we utilised five biomass burning inventories, including the Global Fire Emissions Database (GFED), which is frequently used for model output comparisons due to its extensive historical data record. We employed both the GFED Beta version 5 (GFEDvn5) and version 4.1s (GFEDvn4s) (van der Werf et al., 2017; Chen et al., 2023), alongside the Global Fire Assimilation System version 1.2 (GFASvn1.2) (Kaiser et al., 2012) and the Fire INventory from NCAR version 2.5 (FINNvn2.5) (Wiedinmyer et al., 2023). Additionally, a regional inventory, the Brazilian Biomass Burning Emission Model (3BEM-FRP)(Pereira et al., 2022), was utilised. These inventories are based on three distinct fire products: GFED uses the burnt area (BA) as the base satellite product, FINNvn2.5 uses active fire hotspots (from which the BA is calculated), and GFASvn1.2 and 3BEM-FRP are based on fire radiative power (FRP). GFASvn1.2, FINNvn2.5 and 3BEM-FRP had daily data with a spatial resolution of 0.1° × 0.1°. While GFEDvn5 and GFEDvn4s were downloaded with a monthly resolution and a spatial resolution of 0.25° × 0.25°. All the inventories were resampled to a monthly temporal resolution and a spatial resolution of 0.5° × 0.5° to match the model outputs dimensions in the JULES-INFERNO configuration (Section 2.4).

GFEDvn4s was the first GFED version to consider a correction for small fires using 500m MODIS bands (van der Werf et al., 2017). With GFEDvn5, the efforts for including small fires were reiterated by including a correction based on Landsat or Sentinel-2 observations (Chen et al., 2023). FINNvn2.5 has a special adjustment to calculate BA in forest areas, where hotspots of fire activity are clustered together to overcome low visibility caused by tree canopy interference (Wiedinmyer et al., 2023). GFASvn1.2 uses the MODIS near-real-time FRP product to estimate real-time emissions (Kaiser et al., 2012). 3BEM-FRP



also uses FRP from MODIS observation, applying an adjustment factor to account for fires beyond the spatial and temporal
resolution of the product. This adjustment is based on comparisons with VIIRS and the geostationary satellites GOES and
SEVIRI (Pereira et al., 2022).

The estimation of CO emissions by these inventories varies significantly due to differences in how each one calculates the
amount of burned dry matter (Hua et al., 2024). However, to convert this burned matter into emissions, all inventories rely on
emission factors (EF) $[gkg^{-1}]$ that vary by land use and land cover. The EFs are consistently derived or partially derived from
the studies conducted by Akagi et al. (2011) and Andreae and Merlet (2001). Particularly, the EF for CO shows less variation
across different inventories compared to other compounds (Liu et al., 2020; Hua et al., 2024). GFASvn1.2 and 3BEM-FRP use
a combustion factor to determine the amount of biomass burnt by different levels of energy. They rely on external products,
GFEDvn3.1 for GFASvn1.2 and the Fire Energetics and Emissions Research vn1 (FEERvn1) for 3BEM-FRP (Kaiser et al.,
2012; Pereira et al., 2022).

Annual Land cover is another important input for the inventories. Here, the MODIS MDC12Q1 collection 5.1 or 6 is used.
Specific for Brazil and the Amazon, 3BEM-FRP includes the MapBiomas collection 6, which better captures the deforestation
process in the Amazon and forest formation in northern Cerrado (Mataveli et al., 2023).

## 2.3 TCCO retrievals

We used the TCCO from the Infrared Atmospheric Sounding Interferometer (IASI) by the University of Leicester IASI Re-
trieval Scheme (ULIRS) (Illingworth et al., 2011), and the version 9 level 2 product from the Measurements Of Pollution In The
Troposphere (MOPITT) developed by NASA/LARC/SD/ASDC (2022) (Deeter et al., 2022). The IASI TCCO record used in
this study is between 2014 and 2021 since ULIRS was applied to the instrument on Metop-B (satellite launched in September
2012). IASI has a circular footprint at nadir with a diameter of 12 km, extending to an ellipse of ∼39 km × 20 Km at the edge
of the swath. Due to its wide swath, the global coverage is achieved in 12 hours. To estimate the TCCO, ULIRS used the IASI
band centred on $4.7\mu m$, with absorption ranging from 2040 to 2190 cm$^{-1}$, but because there are other stronger absorbers in
this domain (e.g., H$_2$O, CO$_2$, O$_3$), then only the range 2143 cm$^{-1}$ to 2181 cm$^{-1}$ is utilized. From this, ULIRS uses an optimal
estimation method to determine the CO profile from the measured radiance (Illingworth et al., 2011).

MOPITT is on board Terra and has a horizontal resolution of 22 km ×22 km, a swath of 640 km and a global coverage
every 3-4 days. Three retrieved products are available: TIR-only, NIR-only, and TIR/NIR. For this study, we used the TIR/NIR
product, which has the highest sensitivity in the lower troposphere. MOPITT uses the radiative transfer model recognised as
the MOPITT operational fast-forward model (MOPFAS) and an optimal estimation-based algorithm to retrieve TCCO (Deeter
et al., 2017).

The TCCO products were gridded at 0.5° × 0.5° and at a monthly average resolution co-locating the model and inventories.
Only retrievals with a degree of freedom signal (DOFS) ≥1, cloud fraction ≤ 20% and solar zenith angle <90° were included
in our analysis. This last criterion was established to focus solely on daytime products, ensuring fair comparisons between the
retrieval products.



## 2.4    JULES-ES: ISIMIP3a setup

The JULES-ES configuration for ISIMIP3a was utilised in this study. This configuration is described in Mathison et al. (2023) and functions as an offline land/vegetation model, requiring prescribed atmospheric inputs while having a setup similar to that of the UKESM (Sellar et al., 2019). For this study, we ran the model from 2001 to align with the study period (2004-2021). This JULES-ES configuration has a spatial resolution of $0.5° \times 0.5°$ latitude–longitude grid.

ISIMIP3a includes a core of experiments based on climate-related forcings and direct human forcings (Frieler et al., 2024). For climate-related forcings, the models used four standard observation-based meteorological datasets: GSWP3-W5E5, 20CRv3-W5E5, 20CRv3-ERA5, and 20CRv3 (Frieler et al., 2024). Some datasets are composites of two separate reanalysis datasets: historical and recent. Aligned with our study period, we utilised only the most recent part of the dataset. As a result, we only considered the climate forcing datasets W5E5, ERA5, and 20CRv3. Both GSWP3-W5E5 and 20CRv3-W5E5 relied on W5E5 data from 1979 to 2019 (Frieler et al., 2024). The 20CRv3 dataset ends in 2015, while the ERA5 dataset ends in 2021. In this study, ERA5 is used for the control analysis as this covers the study period (2004 - 2021). The other two datasets (i.e. 20CRv3 and W5E5) were utilised for the flammability experiments described in Section 2.5.

The human-forcing datasets in ISIMIP3a prescribe land use (agricultural and pasture fraction), population density (PD), and nitrogen deposition. For this study, the human development index (HDI) was prescribed to represent socioeconomic factors as suggested by Teixeira et al. (2021), but only for the experimental work (Section 2.5). Since the original version of INFERNO in ISIMIP3a does not prescribe this, HDI=0 was used in the control run.

In this setup, the land component, JULES, contains 13 plant functional types (PFTs) (listed in Table 1), which include four managed and nine natural PFTs (Mathison et al., 2023). The four managed PFTs are C3 and C4 crops (C3Cr and C4Cr) and pastures (C3Pa and C4Pa). JULES-ES also contains four non-vegetation land covers (soil, lake, ice, urban). The PFTs can be globally distributed by simulation within JULES-ES using the dynamic global vegetation model (DGVM) called TRIFFID (Top-down Representation of Interactive Foliage and Flora Including Dynamics), which models the PFTs competition and their biomass (Burton et al., 2019).

### 2.4.1    INFERNO

The INFERNO model developed by Mangeon et al. (2016) uses PFTs as vegetation categories for the estimation of burned area (BA), emitted carbon (EC) and emitted species ($E_x$). To calculate the BA, INFERNO uses total ignition ($I_T$), flammability ($F_{PFT}$) and an average burned area ($\overline{BA_{PFT}}$) as described in Equation 1.

$$BA_{PFT} = I_T \, F_{PFT} \, \overline{BA_{PFT}} \tag{1}$$

Table 1 lists the $\overline{BA_{PFT}}$ used for each PFT in this study. $I_T$ and $F_{PFT}$ behave as probabilistic variables ranging from 0 to 1. In INFERNO, $I_T$ is split into natural ignition ($I_N$) and anthropogenic ignition ($I_A$).

Three ignition methods can be used in the model. The first and simplest is "constant ignition", where $I_N$ and $I_A$ are constant. Here, $I_N$ assumes a multi-year annual mean lightning rate of 2.7 flashes/km$^2$/yr, where 75% are cloud-to-ground, all of which



**Table 1.** JULES-ES's Plant Functional Type (PFT) and their respective Average Burnt Area ($\overline{BA_{PFT}}$) and Emission Factor ($EF_{PFT}$) for INFERNO modelling

| PFT | Short name | $\overline{BA_{PFT}}[km^2 fire^{-1}]$ | $EF_{CO}[gkg^{-1}]$ |
|---|---|---|---|
| Broadleaf deciduous trees | BDT | 0.6 | 93 |
| Tropical broadleaf evergreen trees | BET-Tr | 0.6 | 93 |
| Temperate broadleaf evergreen trees | BET-Te | 0.6 | 89 |
| Needleleaf deciduous trees | NDT | 0.6 | 127 |
| Needleleaf evergreen trees | NET | 0.6 | 89 |
| C3 grass | C3G | 1.4 | 89 |
| C3 crop | C3Cr | 0.2 | 0 |
| C3 pasture | C3Pa | 1.4 | 98 |
| C4 grass | C4G | 1.4 | 63 |
| C4 crop | C4Cr | 0.2 | 0 |
| C4 pasture | C4Pa | 1.4 | 63 |
| Deciduous Shrub | DSh | 1.2 | 89 |
| Evergreen Shrub | ESh | 1.2 | 127 |

provoke $I_N$. The $I_A$ is 1.5 ignitions/km$^2$/month globally, based on GFED estimations (Mangeon et al., 2016). The second
185    ignition method is "varying natural ignition", which uses constant $I_A$ as the first ignition method but varying $I_N$ (i.e., lightning).
The annual seasonality of cloud-to-ground lightning is prescribed. The third method, "varying natural and human ignitions",
uses the same varying $I_N$ as the second method and a varying $I_A$, which depends on prescribed PD and optional HDI (Teixeira
et al., 2021, 2023). The $I_A$, described in Equation 2, uses a distinct anthropogenic influence on ignitions in rural versus urban
areas represented by $k_{(PD)} = 6.8 \times PD^{-0.6}$. In this equation, $\alpha$ is the number of potential ignitions per person per month per
190    km$^2$ with a constant magnitude of 0.03.

$$I_A = k_{(PD)} \, PD\alpha \times (1 - HDI) \tag{2}$$

This third ignition method attempts to include anthropogenic fire suppression, so the fraction of fires not suppressed by
humans ($f_{NS}$) is included for the calculation of $I_T$ in Equation 4. Since HDI is not included in the control model, then HDI=0.

$$f_{NS} = 7.7(0.05 + 0.9 \times e^{-0.05PD}) \times (1 - HDI) \tag{3}$$

195

$$I_T = (I_N + I_A)\frac{f_{NS}}{8.64 \times 10^{10}} \tag{4}$$





For INFERNO, the term $F_{PFT}$ (described in Equation 5) depends on the relative humidity (RH) in %, precipitation rate (R) in mm day$^{-1}$ and temperature in K from the prescribed input meteorological dataset. The land surface model, JULES, provides the inputs of soil moisture content ($\theta$) as a fraction of saturation, and fuel load (leaf carbon and decomposable plant material) [kg m$^{-2}$]. These are used to calculate the fuel load index (FL) and the Goff-Gratch saturation vapour pressure ($\alpha$), further explained in Mangeon et al. (2016).

$$F_{PFT} = \begin{cases} 1 & \text{for} \quad RH < RH_{low} \\ \alpha \frac{RH_{high} - RH}{RH_{high} - RH_{low}} e^{-2R} FL_{PFT}(1-\theta) & \text{for} \quad RH_{low} \leq RH \leq RH_{high} \\ 0 & \text{for} \quad RH > RH_{high} \end{cases} \qquad (5)$$

For Equation 5, RH$_{low}$ = 10 % and RH$_{high}$ = 90 %. Those are used to scale the influence of RH from 0 to 1. Consequently, notice that $F_{PFT}$ ranges from 0 to 1.

After calculating BA, the emitted carbon ($EC_{PFT}$) is calculated based on the available carbon ($C_i$) and the combustion completeness ($CC$) for wood and leaves. This last term describes the minimum and maximum carbon fraction burnt in the fire events and may/may not depend on PFT. For the ISIMIP3a, $CC_{min,leaf} = 0.8$, $CC_{max,leaf} = 1$, $CC_{min,wood} = 0$ and $CC_{max,wood} = 0.4$ regardless of PFT. Equation 6 defines the $EC_{PFT}$.

$$EC_{PFT} = BA_{PFT} \times \sum_{leaf,wood}^{i} (CC_{min,i} + (CC_{max,i} - CC_{min,i})(1-\theta))C_i \qquad (6)$$

The emission of a compoun X (in this case CO) is described by Equation 7, which includes the $EC_{PFT}$ and the EF for compound X. (EF$_X$) varies for different PFT and is listed in Table 1.

$$E_{X,PFT} = EC_{PFT} \, EF_{X,PFT}/[C] \qquad (7)$$

In this equation,[C] describes the dry carbon fraction, which is assumed to be 50% (Mangeon et al., 2016).

## 2.5 Sensitivity experiments on JULES-INFERNO

JULES-INFERNO refers to the coupled interaction of INFERNO fire simulation in JULES, but throughout the manuscript, we refer to this only as INFERNO. We conducted multiple experiments to assess the sensitivity of various processes and parameters controlling simulated fire emissions. We have divided the experiments into seven sub-groups (Table 2): ignitions, flammability, burnt area, combustion completeness, emission factor, feedback and PFTs. The label of the experiments described the subgroup to which they belonged. The experiment names, groups and details are summarised in Table 2. For ignitions, $IT_N$ and $IT_A$ used the different types of I$_T$ provided by INFERNO and used prescribed HDI with a national and subnational dataset (IT-HDI and IT-HDIS), since the control run uses varying natural and human ignitions and HDI=0. Regarding flammability, the control



**Table 2.** Description of the experiments run with INFERNO

| Short name | Impacted process | Description |
|---|---|---|
| Control | | Varing $I_A$ and $I_N$ ignition |
| IT-CST | | Constant $I_A$ and $I_N$ ignition |
| IT-NAT | Ignition | Constant $I_A$ |
| IT-HDI | | Including HDI dataset national resolution |
| IT-HDIs | | Including HDI dataset subnational resolution |
| F-W5E5 | Flammability | Uses W5E5 climate-forcing dataset |
| F-20CR | | Uses 20CRv3 climate-forcing dataset |
| BA-AVG | Burnt area | Average $\overline{BA}$ regardless of PFT |
| BA-RND | | Randomly switched $\overline{BA_{PFT}}$ through PFT |
| CC-VAR | CC | Varying CC based on van Leeuwen et al. (2014) |
| CC-EXT | | Extended CC from 0 to 1 |
| EF-AVG | Emission factor | Average $EF$ regardless of PFT |
| EF-RND | | Randomly switched $EF_{PFT}$ through PFT |
| NO-FDBK | Feedback | Turn off outputs from INFERNO to JULES |
| EC-PFT | PFTs | Turn off outputs from INFERNO to JULES |

experiments used the ERA5 climate-forcing dataset, while F-W5E5 and F-20CR used the W5E5 and 20CRv3 datasets. For the burnt area, emitted carbon, and combustion completeness, the parameters $\overline{BA_{PFT}}$, $CC$, and $EF$ were modified accordingly.

225     In addition, a no-feedback experiment was also conducted (NO-FDBK), which disables the outputs from INFERNO being passed to JULES and TRIFFID (i.e., INFERNO does not contribute to carbon losses and does not influence fire disturbance to the PFTs). A prescribed PFT experiment was conducted (EF-PFT), using an annual resolution PFTs dataset generated by JULES in the ISIMIP3a team (Mathison et al., 2023) based on the work of Harper et al. (2023) and prescribed land use from the Land-Use Harmonisation dataset provided for ISIMIP3 (Volkholz and Ostberg, 2022). This dataset covers the study period only until 2019. The dominant prescribed PFTs through SA according to the dataset are illustrated in Fig. 1a compared with

230 the output from TRIFFID (1b).

    Figure 1a illustrates the dominant PFT for the prescribed dataset, while Fig. 1 b shows the modelled PFT by TRIFFID. The distribution of PFT presents slight changes in North-SA, where the dominant PFT, according to both, is BET-Tr (∼50%). For Mid-SA, BET-Tr is also dominant; however, the modelled PFT distribution also presents a high fraction of BDT (13%), C4G (18%), and soil (15%). BET-Tr for the prescribed dataset is around 50%, while TRIFFID modelled around 27% throughout

235 the study period. Contrary to the other two regions, in South-SA, the soil cover dominates (∼25%), followed by the PFT C3Pa (∼15%) according to both the simulated PFT and the observation-based dataset.



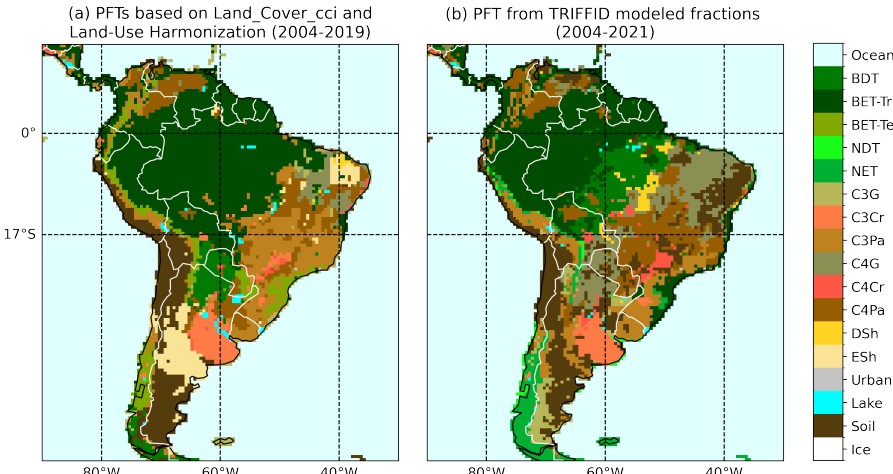

**Figure 1.** Average dominant Plant Functional Types over South America from 2004 to 2021 (a) generated by JULES ISIMIP3a team (Mathison et al., 2023) based on ESA Land Cover Climate Change Initiative (Land_Cover_cci) and the Land-Use Harmonisation datasets and (b) modelled by TRIFFID for this study.

## 2.6 CO emissions variability and comparison

We use the Average Relative Range (ARR) of estimated fire CO emissions across different inventories to quantify the level of variation among them. To calculate the ARR, we first compute the annual range of total CO emission estimates ($y$) for each year $i$ in the study period, defined as the difference between the maximum and minimum values across inventories. These ranges are then normalized by dividing it by the average emission magnitude of the respective year. The resulting values are averaged over the entire study period and multiplied by 100 to express the ARR as a percentage as Equation 8 describes.

$$ARR = \frac{1}{N} \sum_{i}^{N} \frac{max(y_i) - min(y_i)}{\overline{y_i}} \times 100 \tag{8}$$

To assess the significance of comparisons between pairs of CO emissions estimated by different inventories or between an inventory and INFERNO, we employ the non-parametric Mann-Whitney U rank test.

The emissions trend was determined using the Man-Kendall test based on annual CO emissions. We calculate the trend for two periods: the complete study period (2004 to 2021), which is the long-term trend, and the period from 2014 to 2021, which is the short-term trend. The short-term period was chosen based on the availability of the IASI TCCO dataset, which was retrieved from 2014 for the instrument on board Metop-B. We calculate gridded trends, but general assessments are made using regional-scale data.

We utilised the percentage mean bias (MB%) to assess the model's biases related to spatiotemporal variations, seasonal cycles, and trends. As Equation 9 describes, this metric calculates the average difference between a set of values based on the



model (denoted as $x$) and a corresponding set of observational values (denoted as $y$). The observational values are used to scale these differences as the equation shows.

$$MB\% = \frac{1}{N} \sum_{i}^{N} \frac{x_i - y_i}{y_i} \times 100 \tag{9}$$

In Equation 9, $x$ and $y$ refer to different variables depending on the MB% to be calculated. For the spatiotemporal MB%, $x$ and $y$ denote annual CO emissions, while $i$ refers to each year within the 18-year study period. In contrast, for the seasonal cycle MB%, $x$ and $y$ indicate the seasonal cycle amplitude for each year, determined by subtracting the maximum monthly CO emissions during the peak period from the maximum emissions during the non-fire season (based on observations), with $i$ again representing each year. Finally, for the trend MB%, $x$ and $y$ correspond to the trend over a specified range of years, and $i$ outlines the sets of years spanning from (2014, 2021), (2013, 2021), down to (2014-n, 2021), ultimately reaching (2004, 2021). It is important to note that for the trend MB%, $N$ equals 11 (total set of years included).

Additionally, $x$ magnitudes were replaced with the corresponding information from every model run, including control and the experiment proposed in Section 2.5. Similarly, $y$ magnitudes were replaced by the five inventories in this study. So every model run was compared against every inventory.

### 2.6.1 Machine learning for understanding INFERNO CO emission bias

Similar to other studies (Hess et al., 2023; Liu et al., 2022), we employed machine learning (ML) to assess model biases. Specifically, we calculated the annual biases of INFERNO by taking the difference between the total annual emissions simulated by INFERNO and the average annual emissions estimated by the inventories. Our target variable consisted of pixel-scale biases. The primary objective of our analysis was to identify the key factors contributing to these biases and to determine whether the inputs of INFERNO (both prescribed and modelled variables) were sufficient to explain the entire bias. For this analysis, we utilised a gradient-boosting framework implemented using the Python library XGBoost.

We selected 20 inputs for the ML model, comprising prescribed data and JULES outputs used by INFERNO to calculate emissions. These are: population density, lightning flash rates, precipitation rate, relative humidity, temperature, soil moisture, HDI, Wood carbon, leaf carbon and 11 PFTs. From Table 1 all PFTs were considered; however, NDT and NET were merged to a single needle leaf (NT) type, and DSh and ESh were merged to a single shrub (Sh) type. From these, soil moisture, wood carbon, leaf carbon, and PFTs were directly taken from JULES simulations. The other variables were obtained from the original datasets prescribed as inputs to INFERNO. The inputs to the ML model are the gridded datasets resampled to an annual resolution for the study period (2004 to 2021).

The 20 features were compared to remove those that cause redundancy in our predictors and ensure independence between features. To evaluate multicollinearity, we calculated the correlation between pairs of factors and the variance inflation factor (VIF), which describes how much of the variability of a particular feature can be explained by the other features. A VIF lower than 10 is recommended.



The data were randomly split into training (80%) and testing (20%) datasets in a five-fold cross-validation exercise to ensure
the independent performance of the specific training/test sets. The model was evaluated using the coefficient of determination
($R^2$) metric, the Root Mean Square Error (RMSE) and Mean Absolute Error (MAE).

We ran a hyperparameter tuning to select the parameters that lead to the best model performance. This was conducted using a
random search method on each training set in a five-fold cross-validation. We used Sklearn's RandomizedSearchCV with 500 it-
erations (i.e., n_iter). The considered parameters were: max_depth, gamma, reg_lambda, colsample_bytree, min_child_weight,
learning_rate, subsample, n_estimators. For information about these parameters, see Chen et al. (2025).

Since the objective was to identify the key factors contributing to the biases in CO emissions from INFERNO, special
attention was given to the feature contribution methodology. We employed the Shapley additive explanations (SHAP) method,
which assigns an importance value to each feature, known as the SHAP value. This value represents the expected marginal
contribution of a feature and can be either positive or negative. It is calculated by taking the weighted average of all possible
subsets of the selected features in which the specific feature can contribute (Lundberg and Lee, 2017). A SHAP value is
calculated for every predictor in the testing dataset. We calculate the SHAP values using a five-fold cross-validation approach.
For this, we randomly split the dataset into five groups. In each of the five iterations, four of the groups are used as the training
dataset, while the remaining group serves as the testing dataset. This allows us to calculate a complete set of SHAP values
for the whole dataset and calculate a map of contribution. We finished with 18 groups of SHAP values for every pixel that
correspond to the number of annual CO emission biases included in the analysis.

We utilise two additional features to assess the SHAP values: the first feature categorises the pixels into North-SA, Mid-SA,
and South-SA, while the second feature identifies the pixel's location on a map. With this, we can describe SHAP values based
on their geographical location. These extra features were only used after calculating the SHAP values, so they were not used to
train the model. To calculate the dominant feature by pixel, we identify the feature with the largest positive (negative) SHAP
value on pixels with an average positive (negative) CO emission biases. Once we established the most important feature for
each pixel across the years, we calculated the mode to identify which feature consistently contributes the most.

## 3 Results and discussion

### 3.1 Estimated and modelled CO fire emissions in SA

Most of SA's CO fire emissions are concentrated in the Arc of Deforestation region in Mid-SA, as shown by both the in-
ventories and INFERNO (Fig. 2). Here, FINNvn2.5 estimates the highest annual fire CO emissions (70.8 $Tgyr^{-1}$), followed
by GFEDvn5 (37.0 $Tgy^{-1}$). GFEDvn4s, GFASvn1.2, 3BEM-FRP and INFERNO are approximately one-third of FINNvn2.5
emissions in the Arc of Deforestation with 21.1 $Tgyr^{-1}$, 19.0 $Tgyr^{-1}$, 21.2 $Tgyr^{-1}$ and 26.3 $Tgyr^{-1}$, respectively. The
estimations of the inventories on the Arc of Deforestation have an ARR of 157% and represent around 30-80% of the total
annual CO emissions from fires in Mid-SA, listed in Table 3. However, this is lower for INFERNO (23%), despite estimating
similar CO emissions as the inventories for both the deforestation front and Mid-SA. This is in part because INFERNO cannot
accurately reproduce the specific details of the deforestation zone, even though it broadly identifies the area. Many fires in this



region occur on a smaller scale than the INFERNO resolution, even overlooked by MODIS products (1 km resolution) (Liu et al., 2020). The inventories display finer detail due to their five times higher resolution and adjustments for smaller fires, such as those included in GFEDvn4s and GFEDvn5. However, GFEDvn4s demonstrates limitations in identifying emissions on the
eastern side of the Arc of Deforestation in comparison to the updated version, GFEDvn5. According to Teixeira et al. (2021), the INFERNO model overestimates emissions in this area by up to 300% compared to GFEDvn4s. However, GFEDvn4s likely underestimates emissions in this region, as highlighted.

The Mid-SA region also contains the Cerrado ecoregion, a fire-prone ecosystem with a mixture of grasslands, shrublands and forests, where fire frequency ranges from 3 to 6 years (Júnior et al., 2014). This accounts for around 10% to 20% of CO
emissions in Mid-SA. GFEDvn5, FINNvn2.5 and INFERNO estimate the lower contribution of these zones to the total emissions. The inventories, however, agree on estimating CO emissions of around 9.3 $Tgyr^{-1}$ (ARR:39%). INFERNO estimation of 7.5 $Tgyr^{-1}$ is in the interquartile range of the annual estimations from most of the inventories (except FINNvn7.5).

It is clear that GFEDvn5 and FINNvn2.5, as well as INFERNO, present particularly high magnitudes of CO emissions in forest areas (see Figure 1 and 2). This might be related to the higher BA calculated by these inventories. For FINNvn2.5,
only for the forest biome, multiple fire detections in adjacent pixels are assumed to correspond to a large fire (Wiedinmyer et al., 2023). However, using this approach plus VIIRS observations led to overestimating CO emissions in the southern part of the Amazon forest (Wiedinmyer et al., 2023). The version of FINNvn2.5 used in this study (i.e., based only on MODIS fire hotspots) estimates significantly lower CO emissions than the version that includes VIIRS hotspots for SA. On the other hand, GFEDvn5 contains an adjusted BA based on Landsat observations, although this is still a beta version. GFEDvn5 estimations
surpass the BA and carbon emissions estimated by GFEDvn4s (Chen et al., 2023; Qi et al., 2024; Blackford et al., 2024), and consistently surpass the CO emissions estimated by GFEDvn4s in this study (p-value <0.05).

For this study, GFASvn1.2 has the lowest average annual CO emission in Mid-SA, yet with a similar annual regional distribution to 3BEM-FRP and GFEDvn4s (p-value≥0.05), see Table 3. However, Naus et al. (2022) suggest an underestimation of CO emissions from GFASvn1.2 after prescribing the emissions into an atmospheric model and comparing the calculated
TCCO against the TCCO retrieved by MOPITT and IASI (Naus et al., 2022).

With ∼25% of CO annual emissions in SA, North-SA and South-SA also contain hotspots of particularly high CO fire emissions. In North-SA, the fire-prone Llanos ecregion, a mosaic of grasslands and savannas between Colombia and Venezuela, contains around 35% CO fire emissions in the subregion according to most of the inventories, except for 3BEM-FRP, which estimates 58% of the annual emissions for North-SA contributed by the Llanos ecoregion (4.6 $Tgyr^{-1}$). The ARR magnitude
for the Llanos ecoregion, including (not including) 3BEM-FRP is 129% (46%). INFERNO estimates a smaller contribution of 10% (2.1 $Tgyr^{-1}$) for the Llanos ecoregion, but within the inventories range, since the annual estimate of CO emissions for North-SA is significantly higher than for the inventories (p-value <0.05). As in Mid-SA, FINNvn2.5 estimated particularly high emissions on the deforestation front in the north of the Amazon (2.7 $Tgyr^{-1}$), four times higher than other inventories, and two times higher than INFERNO. The ARR of inventories in the Amazon northern deforestation front is 212% (37%) including
(excluding) FINNvn2.5. In South-SA, the Dry and Humid Chaco ecoregion contributes around 37% of CO fire emissions to the region according to most of the inventories, except for 3BEM-FRP, which estimates a contribution of around 54% (9.4





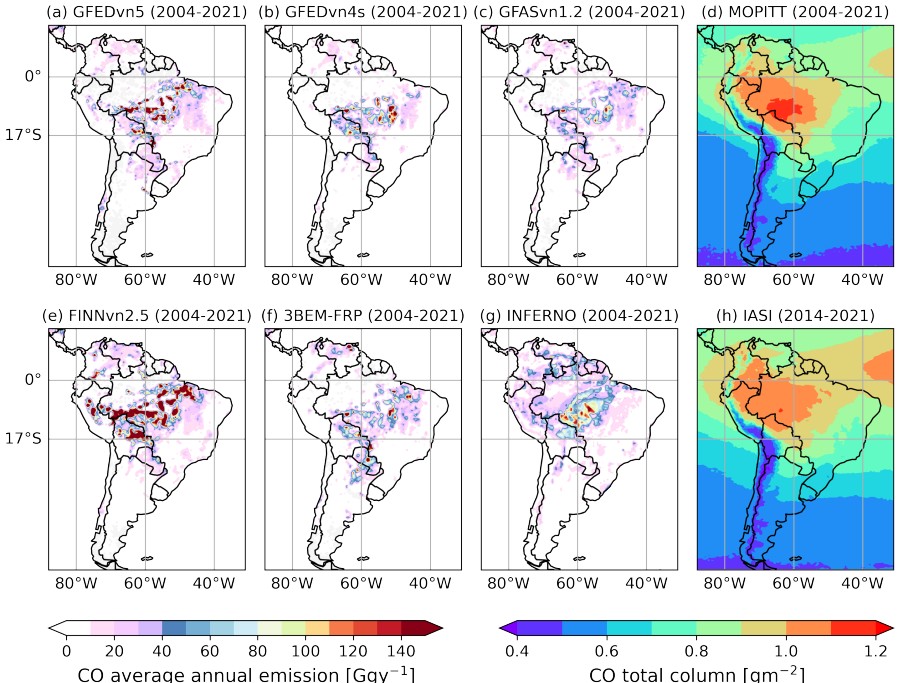

**Figure 2.** Annual mean fire CO emissions for 2004-2021 estimated by (a) GFEDvn5, (b) GFEDvn4s, (c) GFASvn1.2, (e) FINNvn2.5, (f) 3BEM, and modelled by (g) INFERNO and TCCO retrieved from MOPITT (d) and IASI (h). Note that the average TCCO from IASI ranges from 2014 to 2021.

$Tgyr^{-1}$). The high CO emissions from 3BEM-FRP were previously linked to the combustion factor used in the inventory based on FEERvn1.0 (Pereira et al., 2022). The ARR magnitude for the Chaco ecoregion, including (not including) 3BEM-FRP is 132% (66%). In this ecoregion, INFERNO estimates lower annual CO emissions than the inventories (2.3 $Tgy^{-1}$), but

still in the interquartile range of annual emissions estimated by GFEDvn4s, which present significantly lower emissions than the other inventories in this ecoregion (see Fig. 2).

These fire active areas assessed in this study (i.e., Arc of Deforestation, Cerrado, Chaco, Llanos, and Amazon northern deforestation front) explain over 70% of CO emissions in SA; however, they account for around 32% of emissions simulated for INFERNO in the region. This is potentially caused by e.g. the model resolution and simplified process representation, and

the overestimation of emissions over ecoregions, as within the Amazon forest. The inventories, however, describe a broad split of the estimated CO emissions in Mid-SA and North-SA, with ARRs of 138% and 124%, respectively. Without including FINNv2.5, the ARRs fall even below half, 62% and 65%, respectively , suggesting higher agreement (see Table 3). South-SA presents an ARR of 75% (73%) including (no including) FINNvn2.5.

In Fig. 2, the TCCO illustrates how these fire emissions are concentrated in Mid-SA on the east side of the Andean mountain,

where emitted CO accumulate enhanced by the longer lifetime of CO over the Amazon (Lichtig et al., 2024). The accumulation





at the east and north of the Andean mountain range is evidenced in North-SA. Furthermore, the influence of easterly transport of the smoke plume from Africa is clear (Holanda et al., 2020; Lichtig et al., 2024).

### 3.1.1 Intra-annual variability and seasonal cycle of CO fire emissions in SA

SA has a distinct seasonal cycle in fire activity, illustrated in Fig. 3 (b),(c),(d), with high fire activity from August to October
for Mid-SA and South-SA and from January to April for North-SA. The differences between the inventories in the regions studied are significantly enhanced during the peak CO emissions periods, which drives the annual differences discussed above. However, the differences remain consistent through the study period, allowing the inventories to exhibit a high correlation (R > 0.9) for monthly emissions at the regional scale (see Fig. S2). In Mid-SA during the peak period, the estimated CO emissions of FINNvn2.5 exceed all other inventories and INFERNO (p-value≤0.05), while GFASvn1.2 has the lowest average
CO emissions, yet this is not significantly less than GFEDvn4a and 3BEM-FRP (p-value>0.05). During the peak fire period in North-SA, FINNvn2.5 estimates the largest emissions among inventories, probably due to the larger burned area assumed for forest cover. 3BEM-FRP has the next highest emissions, which are also higher than GFEDvn5, GFASvn1.2 and GFEDvn4s (p-value<0.05). These last two inventories demonstrated a similar monthly regionally accumulated distribution (see Fig. 3) of CO emissions and lower magnitudes than GFEDvn5. For South-SA, the peak CO emissions also showed a significant
difference between pairs of inventories. Here, 3BEM-FRP had the highest magnitudes (p-value≤0.05), while GFEDvn4s had the lowest (p-value≤0.05) relative to the other inventories. GFEDvn5 is greater than GFASvn1.2 and FINNvn2.5, while these had a similar distributions.

Figures 3 (b) and (d) show that INFERNO inaccurately represents the seasonal cycle in both North-SA and South-SA; However, its representation in Mid-SA is more consistent with the inventories. For North-SA, the peak period of the fire
activity is well represented, but the model generates a second peak slightly higher than the first one, centred on October. This might be related to the relatively high contribution of gross primary productivity (GPP) to fire activity in INFERNO. While the first simulated peak of emissions in the year appears to be driven by high flammability, the second follows GPP variability, with relative average flammability and precipitation conditions (see Fig. S3). Furthermore, the representation of the seasonal cycle of precipitation in this region in particular may be deficient due to ERA5 limited representation of the Intertropical Convergence
Zone (ITCZ) (Lavers et al., 2022). In North-SA, INFERNO estimated CO emissions are higher than most inventories (p-value ≤ 0.05), particularly outside the peak fire periods.

In contrast to North-SA, INFERNO's CO emissions in South-SA at the observed peak have lower average values than all inventories; in fact, the peak is barely represented. However, the estimated emissions are higher than those in the inventories from November to February. In this period, both simulated flammability and GPP contribute to high fire activity, despite being
the peak of precipitation in the east of the Andean Mountains (Grimm) (see Fig. S3). Although fire activity in this arid ecoregion is highly susceptible to precipitation accumulation which enhanced GPP, it tends to exhibit a delay following the precipitation peak (San Martín et al., 2023).

The TCCO in Fig. 3b-d follows the emission season cycle, showing a rapid increase from August to September in Mid-SA, with a slower decrease corresponding to the long lifetime of CO in the atmosphere. The TCCO in both North-SA and Mid-SA





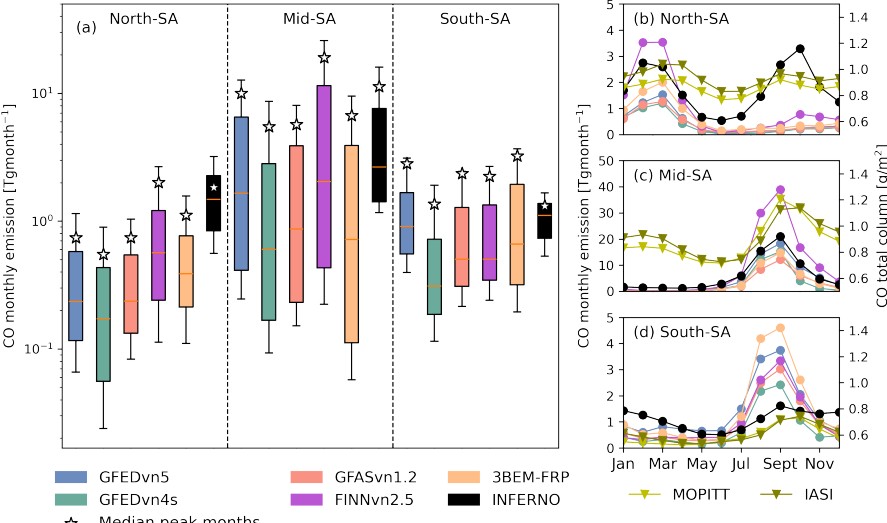

**Figure 3.** (a) Distribution of monthly regionally accumulated CO fire emissions and seasonal cycle for (b) North-SA, (c) Mid-SA and (d) South-SA and their respective mean TCCO seasonal cycle in the period 2004-2021 (2014-2021 for IASI). The retrieved TCCO is represented only in panels (b), (c) and (d), and their magnitudes are read from the right y-axis; those cycles are marked with an inverted triangle. Panels (b), (c) and (d) have different ranges for the left y-axis, while panel (a) has a logarithmic scale for the y-axis. The stars in (a) describe the mean emissions for the fire activity peak for each region that are: from January to April for North-SA, July to October for Mid-SA and August to October for South-SA.

presents a bimodal season, which does not mirror the region's fire CO emissions, but rather evidences transport throughout hemispheres, as well as from Africa to SA.

### 3.1.2 Trends of CO fire emissions in SA

From 2004 to 2021, Mid-SA experienced an average annual decrease of $\sim$2.8%yr$^{-1}$ in CO emissions according to the inventories, with a significant trend observed for GFASvn1.2 (see Table 3). The INFERNO model aligns with the inventories, indicating a negative trend of 1.8% yr$^{-1}$; however, it does not represent a significant decrease in CO emissions in the Arc of Deforestation. In North-SA, both the inventories and INFERNO agree on an increase in CO emissions, with a positive trend ranging from 0.7% yr$^{-1}$ to 9.3% yr$^{-1}$. This is only significant for 3BEM-FRP, which estimates the highest rise in CO emissions. In South-SA, the calculated trend was not significant for the inventories, and there was some disagreement among them regarding the direction of the trend. Nonetheless, most inventories and INFERNO suggested a negative trend of approximately -1.0%yr$^{-1}$.

The observed trends in CO emissions aligned with the observed reduction in BA of around 2%yr$^{-1}$ and 1%yr$^{-1}$ for Mid-SA/South-SA and North-SA between 2001 and 2020 (Chen et al., 2023). However, as for CO emissions, the BA and the carbon emissions did not decrease significantly across all SA (Chen et al., 2023; Aragão and Shimabukuro, 2010; Chen et al., 2013).





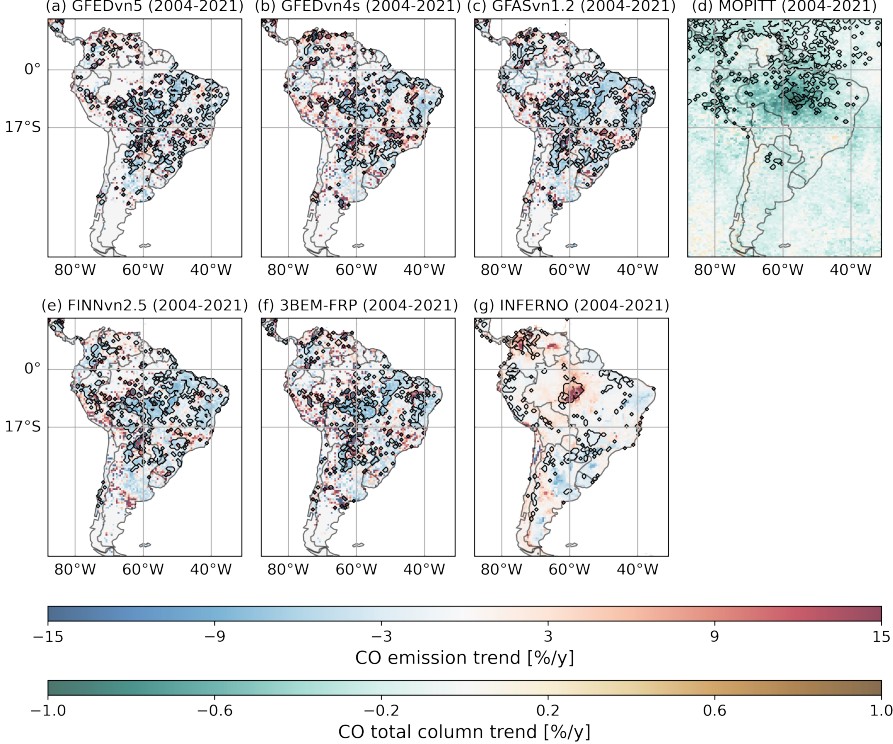

**Figure 4.** Long-term (2004-2021) fire CO emission trend estimation [%/y] based on (a) GFEDvn5, (b) GFEDvn4s, (c) GFASvn1.2, (e) FINNvn2.5, (f) 3BEM-FRP, and (g) INFERNO and TCCO trend estimated based on retrievals from (d) MOPITT. The areas enclosed by the black contour represent zones with a significant trend in CO emissions (p-value≤ 0.05)

As Figure 4 shows, although a large fraction of SA presents a negative trend, there are also areas of positive trends in CO
emissions. The inventories agree with the significant reduction of CO emissions in the Arc of Deforestation, particularly in the states of Pará and Mato Grosso in Brazil, east and south of the Amazon. The emission also decreases significantly in the Caatinga ecoregion in the eastern part of Brazil. In contrast, the northeastern part of the Cerrado, in the large agriculture frontier recognised as MAranhão, TOcantins, PIauí and BAhia (MATOPIBA) presented a positive trend in CO emissions that aligns with an observed increase of BA in part of this frontier (Milare et al., 2024; Pope et al., 2020). Most inventories also agreed
with an increase in CO emissions in the northern coastal region of North-SA and South-SA around the Río Negro province of Argentina, in the middle of the latitude range of the country.

Although contrary to the inventories, INFERNO show an increase in CO emissions in the Arc of Deforestation; it captures the decrease in emissions in the Caatinga region and the slight increase in emissions in northern Colombia and Venezuela and the south in Argentina. With this, INFERNO and JULES simulations suggest some underpinning factors driving the observed
trend. For Caatinga, INFERNO identified decreased ignitions and carbon availability (See Fig. S5). On the contrary, more





wood carbon and increasing ignition were simulated in Argentina (Figs. S4 and S7). For the northern part of North-SA and the Arc of Deforestation, INFERNO describes more flammable conditions.

In support of the general decreasing CO emissions trend among inventories, the MOPITT TCCO identifies a general decreasing trend of 2%yr$^{-1}$, particularly significant in the Amazon and Cerrado ecoregions (Fig. 4 (d)). This has previously been

observed for the period 2003 to 2018, where TCCO evidenced decreasing magnitudes, especially in forested areas (Naus et al., 2022; Deeter et al., 2018), where smouldering ignition dominates, and CO emission factors are high (Deeter et al., 2018). Naus et al. (2022) found a high correlation between the decline in TCCO and a decrease in deforestation enhanced by law enforcement policies in Brazil. In contrast, the increase of CO emissions observed in the north of North-SA and in the northeast of Cerrado was not reproduced by the change in TCCO. Nonetheless, observations suggest an increase in NO$_x$ emissions in the

northeast of the Cerrado (2005-2016) (Pope et al., 2020). The increasing emissions from these biomes, with relatively low CO emissions, might be offset by the surrounding decrease in CO, while the relatively high rate of NO$_x$ emissions is substantial.

**Table 3.** Estimated CO emissions magnitude, seasonal cycle (SC) and trend summary

| Item | GFEDvn5 | GFEDvn4s | GFASvn1.2 | FINNvn2.5 | 3BEM-FRP | INFERNO |
|---|---|---|---|---|---|---|
| | | | North-SA | | | |
| CO emission [$Tgy^{-1}$] | 5.5 (2.0) | 4.4 (1.8) | 5.0 (1.3) | 13.1 (4.4) | 7.9 (2.9) | 21.0 (3.6) |
| SC amplitude [$Tgy^{-1}$] | 1.3 (1.1) | 1.1 (1.0) | 1.1 (0.7) | 3.4 (2.5) | 1.7 (1.2) | -0.2 (2.1) |
| Trend 2004-2021 [% yr$^{-1}$] | 2.9 | 3.3 | 1.4 | 0.7 | **9.3** | 0.8 |
| Trend 2014-2021 [% yr$^{-1}$] | 0.4 | -1.1 | -0.9 | 5.9 | 0.5 | -2.4 |
| | | | Mid-SA | | | |
| CO emission [$Tgy^{-1}$] | 56.6 (23.1) | 37.2 (20.3) | 35.2 (17.1) | 108.4 (53.0) | 39.9 (19.5) | 70.2 (15.8) |
| SC amplitude [$Tgy^{-1}$] | 13.5 (7.8) | 14.8 (9.8) | 9.0 (6.3) | 30.7 (18.9) | 12.6 (9.0) | 17.0 (6.9) |
| Trend 2004-2021 [% yr$^{-1}$] | **-3.4** | -1.8 | **-3.5** | **-3.5** | -1.9 | -1.8 |
| Trend 2014-2021 [% yr$^{-1}$] | 0.5 | 1.1 | -0.9 | 4.9 | 2.4 | 3.0 |
| | | | South-SA | | | |
| CO emission [$Tgy^{-1}$] | 16.8 (4.9) | 8.7 (4.0) | 11.8 (2.7) | 12.5 (3.9) | 17.4 (6.4) | 13.1 (1.2) |
| SC amplitude [$Tgy^{-1}$] | 2.4 (1.4) | 1.9 (1.6) | 2.1 (1.0) | 2.5 (1.5) | 3.5 (2.5) | 0.1 (0.3) |
| Trend 2004-2021 [% yr$^{-1}$] | -2.0 | 0.2 | -1.2 | -2.4 | 0.0 | **-0.8** |
| Trend 2014-2021 [% yr$^{-1}$] | **38.2** | **192.9** | **14.4** | **25.06** | **33.5** | **2.3** |

Note: The SC amplitude was calculated by subtracting the maximum monthly CO emissions during the peak period from the maximum emissions during the non-fire season. The peak period was determined as from January to April for North-SA, July to October for Mid-SA and August to October for South-SA.

In the short term, from 2014 to 2021, the trend of CO emissions changed to positive for Mid-SA and South-SA according to inventories. This for Mid-SA is probably partially explained by the recent increase in deforestation in the Brazil portion of the Amazon, which for 2019 rose by ∼80% after the easing of regulations, which decreased fines and reported infractions

against flora by more than 50% despite the increments (Gatti et al., 2023). In Mid-SA, the CO emission trend switched to ∼



$1.6\%yr^{-1}$ according to four of five inventories, mainly due to anthropogenic activities exacerbated by the extreme drought period 2019-2022 (Geirinhas et al., 2023). In 2019, according to the inventories, this region experienced one of the years with the most fire activity in the recent period. Mid-SA CO emissions were around 116% higher than the average of the previous five years. This increase aligns with a fivefold rise in deforestation compared to the average during the same period across three

Brazilian states surrounding the Amazon (Silveira et al., 2020), which is consistent with the increase in carbon emissions and AOD (Gatti et al., 2023; Yuan et al., 2022). Most of the CO emissions in 2019 came from deforestation fires (Andela et al., 2022).

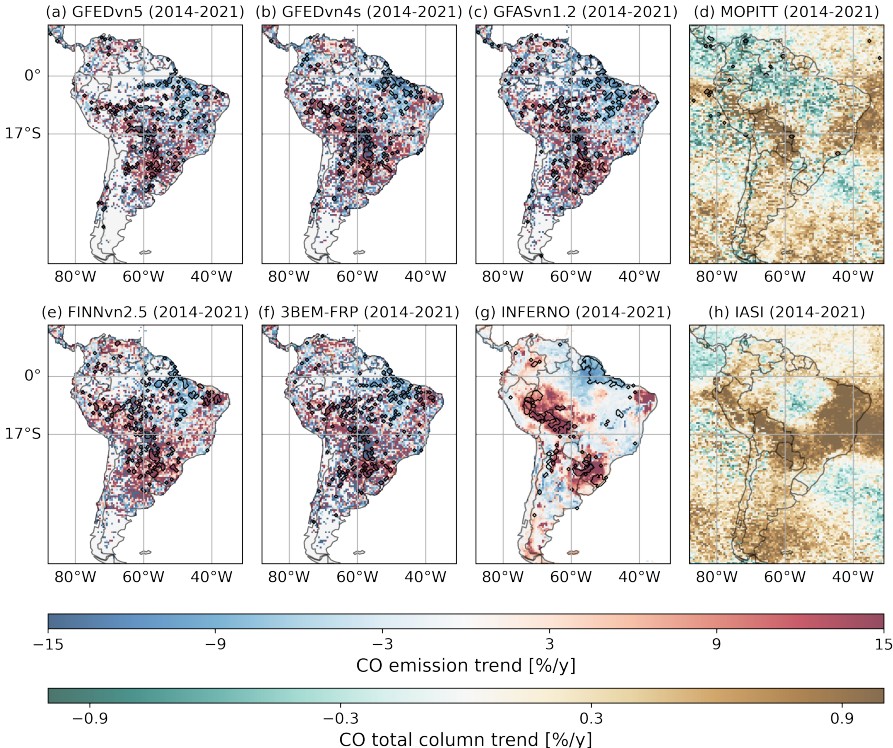

**Figure 5.** Short term (2014-2021) fire CO emission trend estimated based on (a) GFEDvn5, (b) GFEDvn4s, (c) GFASvn1.2, (e) FINNvn2.5, (f) 3BEM-FRP, and (g) INFERNO and TCCO trend estimated based on retrieved from (d) MOPITT and (h) IASI. The areas enclosed by the black contour represent zones with a significant trend in CO emissions (p values≤ 0.05)

In the short term, South-SA presents the strongest trends in CO emissions, with a dominant increasing trend of ∼61.0 $\%yr^{-1}$. Significant trends were found for all the inventories studied. Here, the increased CO emissions were concentrated in

the southern parts of Paraguay and Brazil and the northern parts of Argentina according to GFEDvn5, GFEDvn4s and 3BEM-FRP, which estimate a trend over $85\%yr^{-1}$ in this area. This region encompasses the ecoregions of Humid Chaco, Pantanal, Alto Paraná, and Araucaria. The region experienced unprecedented fire activity in 2020, resulting in emissions that were more than 208% higher than the average for the previous five years. According to Geirinhas et al. (2023), the 2019-2021 drought



period was characterised by an unprecedented soil drought triggered by large-scale interannual forcing, particularly La Niña phase of the El Niño–Southern Oscillation (ENSO), combined with the negative phase of the Pacific Decadal Oscillation. This was additionally likely enhanced by the lack of warm and humid air transported from the Amazon, derived from deforestation (Marengo et al., 2021) and the agricultural expansion in the region (Baumann et al., 2017). Aligned to this, INFERNO describes a significant increase in flammability of around 10% $yr^{-1}$ in this region of the SA low-level jet and the low Chaco, where humid air from the Amazon forest is transported to South-SA (see Fig. S5). The increased fire emissions along the eastern part of the Andean Mountains in Mid-SA also seem to be caused by the same phenomenon. Some inventories also show a propagated pattern in the east of Brazil, reaching the Caatinga. INFERNO does not represent this pattern, although the model represents increased CO emissions in Caatinga, derived from higher ignitions and available biomass.

For North-SA, three out of five inventories suggest an increasing CO emission of around 1%/yr$^{-1}$ in the short-term. This region was also marked by high fire activity in 2020, which caused CO emissions to increase by ~120% for this year compared with the average of the previous five years. For Colombia, this, in addition to the particular dry conditions from September 2019 to March 2020 (Gomes et al., 2021), is also associated with the post-conflict transition. After the peace agreement, when the land occupied by FARC (the Revolutionary Armed Forces of Colombia) was suddenly released, unruled, provoking "uncontrolled" exploitation of natural resources and causing deforestation and fire ignition (Amador-Jiménez et al., 2020). These 2020 CO emissions were around 86% of the CO emissions in 2016, when the fire season was prolonged due to the influence of the El Niño phase of ENSO. Here, as in other studies, INFERNO evidenced the increase in fire activity for this year (Fonseca et al., 2017; Burton et al., 2020). However, INFERNO disagrees with most inventories for North-SA, indicating a negative but insignificant trend in CO emissions. This discrepancy may be due to a model bias in the simulated CO emissions for 2016, which was estimated to be significantly larger than in the subsequent years. However, the inventories that describe the same emissions rate in 2020 do not support this estimate (see Fig. S4). In the short term, INFERNO underestimated the observed increase in CO emissions in northern Venezuela.

For the short term, satellite retrievals support the finding with a positive trend of TCCO through the SA low-level jet along the eastern and central Andes (~1%yr$^{-1}$). This is particularly clear in the TCCO retrieved from IASI, which has more data available and is more sensitive to changes in the upper troposphere. An increase in CO emissions is also observed in the eastern part of SA within the same latitudinal range, which can be attributed to emissions in the Caatinga region. As the inventories with CO emissions, the retrieval products disagree on the direction of the trend of TCCO for North-SA; still, none of the estimates showed a significant trend.

## 3.2 Sensitivity experiments using INFERNO

Due to the likely overestimation of CO emissions from FINNvn2.5 in Mid-SA and North-SA compared to other inventories, and the resulting increase in disagreement, we only used GFEDvn4s, GFEDvn5, GFASvn1.2, and 3BEM-FRP to compare the experiments and calculate the MB% in this section. Here, the INFERNO run assessed in the previous sections is referred to as the control experiment.





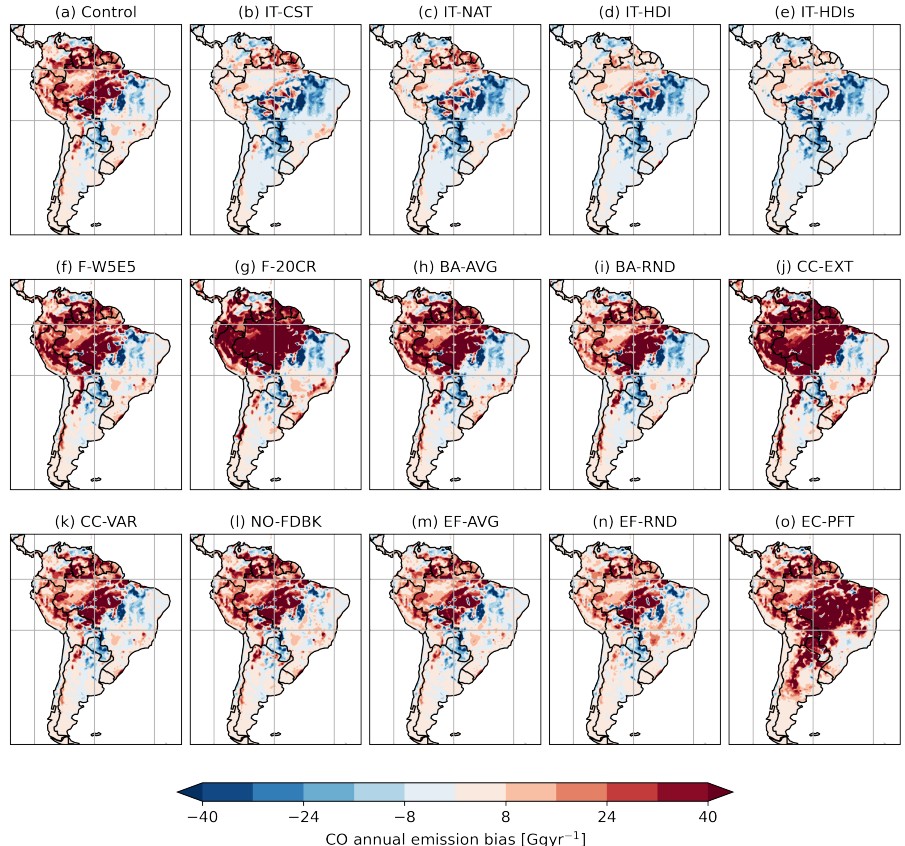

**Figure 6.** INFERNO's experiments CO emissions biases (i.e. difference between CO emissions from INFERNO and the mean of GFEDvn4s, GFEDvn5, GFASvn1.2 and 3BEM-FRP). The control run is the same configuration assessed in Section 3.1. The experiment description can be found in Table 2

The control run shows positive (negative) biases against the average values of the selected inventories over BDT and BET-Tr (C4G) dominated lands based on the PFTs modelled by TRIFFID (see Figs 1 and 6 a). The overestimation of emissions is offset only by the ignition experiments (Fig. 6 b-e), which reduce CO emissions through the territory in about 60%, increasing the underestimation of emissions in the northern part of South-SA and eastern Mid-SA, where most C4G-dominated lands are located (see Fig. S6). The low emissions rate of these experiments, particularly the one related to the addition of the HDI (i.e. IT-HID and IT-HDIs), produces the underestimation of the seasonal cycle amplitude on the three subregions, as Fig. 7 illustrates. The seasonal cycles estimated by the experiments were lower than any estimation from the included inventories and yet produced a lower absolute MB% in Mid-SA (53.4%) than for the control run (96.5%), whose emissions peak is significantly higher than the estimations of GFEDvn4s, GFEDvn5, GFASvn1.2 and 3BEM-FRP. In North-SA, the CO emissions estimated by the ignition experiments, particularly for IT-HDI and IT-HDIs reduced spatiotemporal MB% since the control run presented overestimations.



Both HDI experiments (i.e. IT-HID and IT-HDIs) show the highest level of spatiotemporal agreement with GFEDvn4s in SA (See Fig. S7). Accounting for socioeconomic factors by including the HDI has demonstrated better performance in SA

and various other regions compared to GFEDvn4s (Teixeira et al., 2021). However, the spatiotemporal comparison against GFEDvn5 suggests an increment in the absolute MB% for Mid-SA and South-SA. The spatiotemporal MB% for both HDI experiments in North-SA is lower than the control experiment against all the inventories (Figure S6). The trend described by IT-HDI and IT-HDIs in North-SA also shows a lower MB% than the control experiment, as shown in Fig. 8. The better performance is evidenced by the long-term negative trend of CO emission over the centre of Colombia and the increase of

emission in the north of Colombia and part of Venezuela (see Figs. S7 and S8). Using subnational scale HDI (IT-HDIs), rather than national HDI (IT-HDI), did not improve the performance of INFERNO. On the contrary, in North-SA, the performance on trend MB% of IT-HDIs (-81%) was significantly poorer than that of IT-HDI (-27%).

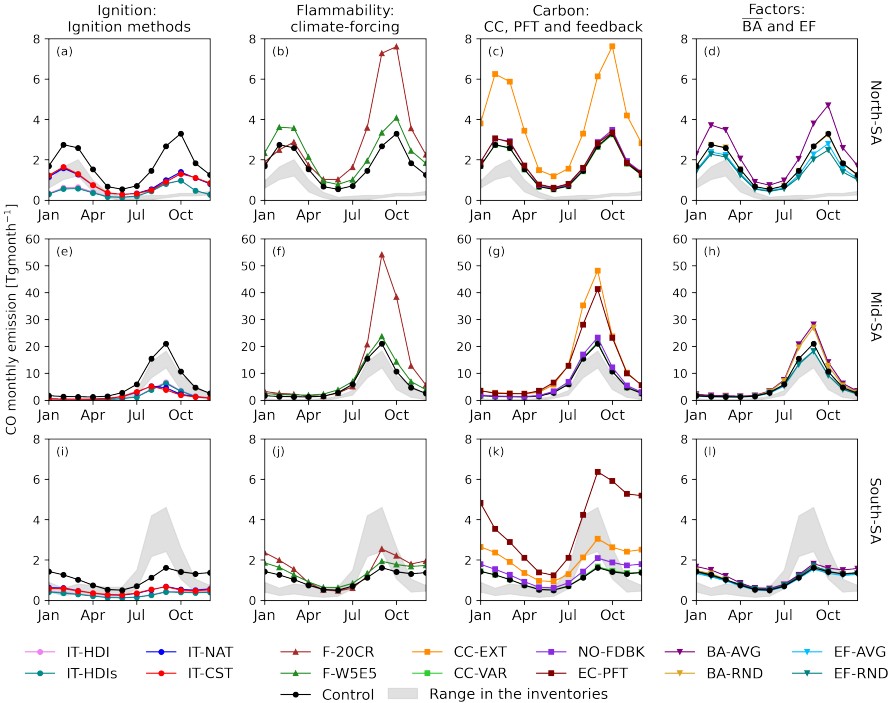

**Figure 7.** CO emissions seasonal cycle modelled by the different INFERNO experiments in North-SA (a-d), Mid-SA (e-h) and South-SA (i-l). The experiments are grouped by type. The inventories range presented in the shaded region omit FINNvn2.5.

Although the Ignition experiment with constant anthropogenic ignition, IT-NAT and IT-CST, performed close to the HDI experiments (i.e. IT-HDI and IT-HDIs) (see Figs. 7 and 8), they particularly differentiated in the trend estimation in Mid-

SA and South-SA. The two experiments described a more negative trend than IT-HDI, IT-HDIS and the control experiment, which reduced biases. However, the strong tendency identified by IT-CST and IT-NAT misses some of the details identified by the control experiments and described in Section 3.1.2. For instance, the trends calculated for Argentina and Caatinga in



the long-term and short-term, respectively (see Fig. S8 and Fig. S9). This exhibits the importance of anthropogenic ignition in this region, which INFERNO can partially describe. Furthermore, IT-NAT and IT-CST have similar CO emissions, even when

IT-NAT have varying natural ignition, which is evidence that the estimated CO emissions have a low sensitivity to the natural ignition variation by the lightning annual cycle, as was described previously for burned area simulation (Burton et al., 2022).

In contrast to the ignition experiments, the experiments using other climatic datasets for simulating flammability (i.e. F-20CR and F-W5E5) do not reduce MB% against the inventories compared to the control experiment run in any of the subregions when evaluated spatiotemporally (see Figs. 6 f-g and 8). Both datasets 20CRvn3 and W5E5 produced higher CO emissions than the

control experiment (i.e., using ERA5), particularly in the BDT and BET-Tr dominant land cover types. The higher monthly estimates of INFERNO based on these two climatologies provide a better representation of emission amplitude in South-SA (see Fig. 7.i), reaching the emissions range of the inventories at their peak, although still high during the non-fire season. In North-SA, the incorrect representation of the seasonal emissions cycle was extenuated, with an incorrect increase in emissions in the non-fire season, as illustrated in Fig. 7.b. This experiment led to the most noticeable changes in the shape of the seasonal

cycle (see Fig. 7b,f,j); however, this is still misrepresented for South-SA and North-SA. This result indicates a systematic bias affecting the simulation of the seasonal cycle, likely because a variable outside the experiments conducted (e.g. GPP), since the simulations only alter the magnitudes of the marked season (see Fig. 7).

The F-W5E5 experiment (i.e., using the W5E5 dataset) represents better trends in the three regions in its shorter run period (i.e. 2004 - 2019) as Fig. 8 shows. In fact, F-W5E5 long-term data show a more accurate decrease in CO emissions in Bolivia

and an increase in CO emissions in Río Negro, Argentina (See Fig. S8).

Changing the PFT (i.e. EC-PFT) was the only experiment that consistently switched the negative spatiotemporal MB% over the TRIFFID C4G-dominated land into positive (see Fig. 6o). This is since the prescribed PFT exhibits rather dominant BET-Tr and BDT, as Fig. 1 illustrates. In Mid-SA and South-SA, because of the land cover change, the EC-PFT calculates significant changes in the seasonal cycle amplitude, which were consistently higher on the emissions peak exceeding the inventory range

(see Fig. 7). In North-SA, however, the changed PFT did not significantly affect the calculated MB% (see Fig. 8 a). For South-SA, the EC-PFT closely describes the trend observed by the inventories (trend bias ∼12%, see Fig. 8 c).

Since the PFTs were prescribed, the EC-PFT did not include feedbacks, like the NO-FDBK experiment. The NO-FDBK experiment indicated that, on average, not including fire feedback in the land model produces 8% higher CO emissions than including it. This can cause a spatiotemporal overestimation of around 50%. Hence, the exclusion of feedbacks could lead to

an increase in biases of EC-PFT emissions.

As expected, due to the contribution of emissions in BDT and BET-Tr, which exhibit a relatively low $\overline{BA}$ and high $EF_{CO}$, the selection of a random or average scheme (i.e., BA-RND, BA-AVG, EF-RND, EF-AVG) consistently increases (reduces) emissions led by higher (lower) $\overline{BA}$ ($EF_{CO}$). Similarly, by extending both wood carbon CC and leaf CC in CC-EXT, the model increases emissions, weighting the now higher potential combustion of wood over the lower combustion of leaf. These

experiments also described the influence of the PFT in BA, EF, and CC by comparing the control experiments with BA-AVG, EF-AVG, and CC-VAR. The influence of the PFT is distributed as BA>EF>CC, according to the absolute spatiotemporal MB% of 30%, 10% and 1% against the control run. Notice that CC-EXT managed a large change (111%) in CO emissions,



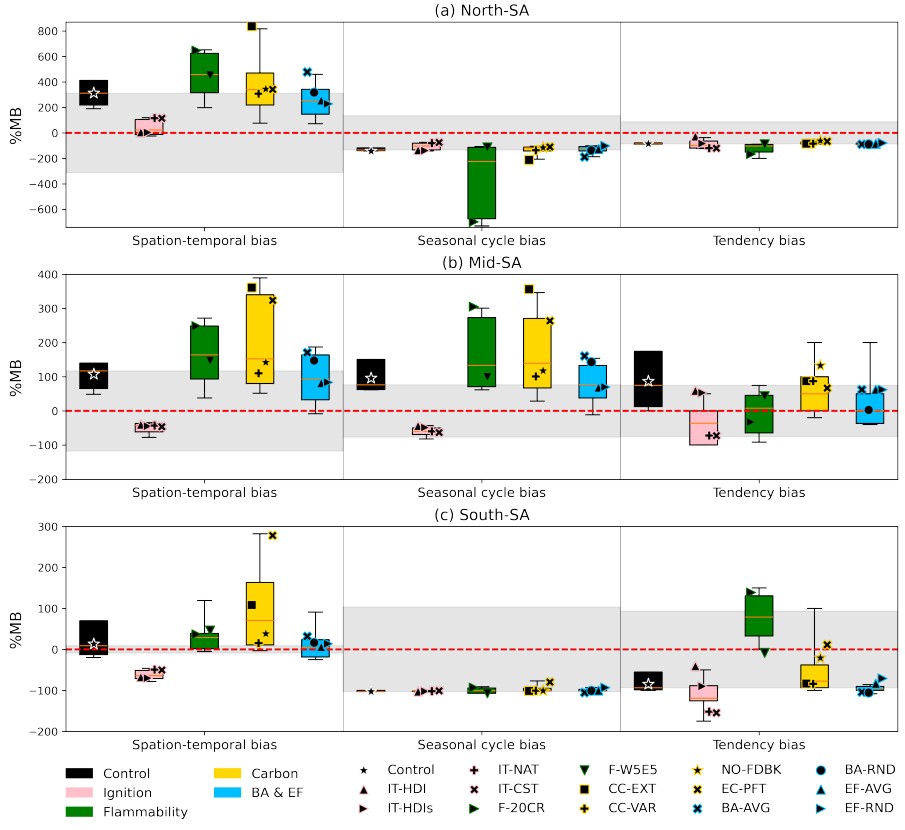

**Figure 8.** Percentage mean bias (MB%) of spatiotemporal, seasonal cycle and trend of CO emissions modelled in the different INFERNO experiments against the studied inventories. The boxes include the comparison of each experiment against the five inventories, with the marker describing the median value of the comparison without including FINNvn2.5. The shaded region in the background represents the absolute median MB% of the control run in each subplot, the reason why it goes from zero to the $\pm$ control's median magnitude. A red line highlights the zero MB%. Note that every subplot has a different y-axis to facilitate visualisation.

but an extreme and hypothetical range was used for this experiment. Contrary to the flammability and ignition experiment, this experiment did not drastically change the INFERNO MB% direction. CC-EXT, however, drastically increases emissions at the

same level or higher than any other experiment conducted here. It represented the sensitivity experiment with perturbed input values to the INFERNO model which were far outside the typical range used for these factors. In contrast, the intent to use more accurate CC (CC-VAR), which depends on PFT, did not show a significant change in any assessment.

In general, including only experiments with realistic values/ranges (i.e. excluding CC-EXT), the Ignition, flammability and PFT sensitivity experiments resulted in the largest changes compared to the control experiment. This is represented in

the average absolute MB% of 115%, 65% and 47% for PFT, ignition and flammability in the spatiotemporal assessment. Flammability presented the largest changes to the seasonal cycles (MB% = 116%), followed by PFTs (MB% = 88%) and then




ignition (MB% = 46%). The changes on the trend were led by PFTs (MB% = 167%), followed by flammability (MB% = 158%) and ignition (MB% = 142%).

## 3.3 Understanding of INFERNO CO emission biases through application of ML

As in the previous section, only the inventories GFEDvn4s, GFEDvn5, GFASvn1.2, and 3BEM-FRP were utilised for the machine learning approach. The XGBoost model's target was the bias of the CO emissions estimated from INFERNO when compared to the average emissions from the selected inventories. After evaluating the features, 14 inputs from INFERNO were included in the final model: 10 PFTs (BDT, BER-Te, NT, C3G, C3Cr, C3Pa, C4G, C4Cr, C4Pa, Sh), soil moisture, lightning, population, and HDI. None of the selected features exhibited a correlation greater than 0.6 with any other feature (see Fig. S10), and the VIFs for these features were below 10. In particular, soil moisture covaries with multiple variables with which its correlation is high, such as relative humidity (R = 0.79), leaf carbon (R = 0.73), wood carbon (R = 0.70), BET-Tr (R = 0.70), and precipitation (R = 0.7). Therefore, these other features were not included directly, but were represented by soil moisture, since soil moisture highly depends on precipitation and is a key variable for GPP, which in turn affects leaf and wood carbon that favour PFTs as BET-Tr.

With the best parameters identified through hyperparameter tuning, the ML model trained using 5-fold cross-validation yielded an $R^2$ value between 0.62 and 0.68 (average 0.64), an RMSE ranging from 18.8 Ggyr$^{-1}$ to 21.3 Ggyr$^{-1}$ (mean of 20.4 Ggyr$^{-1}$), and a MAE ranging from 7.8 Ggyr$^{-1}$ to 8.2 Ggyr$^{-1}$ (mean of 8.0 Ggyr$^{-1}$). Therefore, the ML model is able to explain around 64% of the biases with the available data and data accuracy level. Capturing finer temporal and spatial resolution interactions within the input data can also contribute to reducing the error, since we are using annual datasets. The ML model has particular difficulties addressing negative biases (see Figure S11). This suggests that there are structural and/or parametric deficiencies within INFERNO that particularly limit its ability to represent different fire process patterns through SA. This finding is consistent with Section 2.5, where none of the experiments manage to represent a consistent low MB% compared to the control model for all studied subregions.

Figure 9 presents the feature contributions on two levels: subregional (a) and pixel-by-pixel (b). Since SHAP values can be both positive and negative, we utilised the absolute SHAP values for subregional assessment and the larger positive (or negative) SHAP values to identify areas with an average positive (or negative) bias. According to the SHAP values and consistent with Section 2.5, the BDT fraction (which correlates with BET-Te R=0.6) is the feature of most importance in the three subregions, as Fig. 9.a shows. The higher the BDT fraction, the higher the SHAP values (R = 0.80), and vice versa. This suggests that low values of BDT modelled by TRIFFID also contribute to modelling the lower and/or negative bias of INFERNO CO emissions. BDT is the dominant feature of importance where INFERNO overestimates CO emissions in the Amazon rainforest (see Fig. 9.b). While Teixeira et al. (2021) suggested that an overestimation of tree cover might be a potential driver for the overestimation of emissions in this area, our study shows that even with a lower fraction of tree cover compared to the ESA land-cover based PFTs, we still observe an overestimation of emissions. This indicates that, in addition to the fraction of tree cover, there is also a lack of representation of the fire dynamics affecting these PFTs. From this and based on the observed overestimation of emissions extended in the Amazon forest, we consider that a more accurate representation of fire dynamics





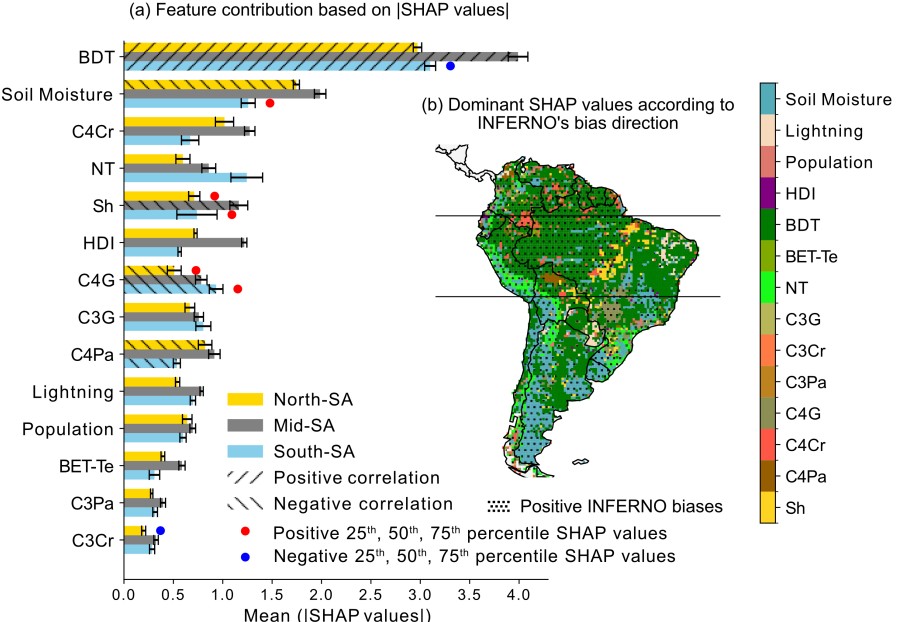

**Figure 9.** (a) Feature contribution of the XGBoost model using absolute SHAP values and (b) map of dominant features based on the largest positive (negative) SHAP values for pixels where INFERNO's average CO emission biases are positive (negative). In (a), the hatch marks describe the correlation between the SHAP values and the annual average magnitude of each feature; only |R| > 0.7 is displayed. The red (blue) filled circles in front of the bars describe when the complete interquartile range (IQR) of the SHAP values in a specific subregion is positive (negative). In (b), the dotted areas indicate when INFERNO's bias is on average positive, while the remaining area shows when the bias is on average negative.

can be achieved by considering landscape fragmentation, representing both forest vulnerability (see Silva-Junior et al. (2022)) and land management (see Andela et al. (2017)), can lead to improved calculation of burned area and fire suppression effect.

In contrast to the Amazon forest, the BDT contribution in the east of Mid-SA and South-SA is negative when CO emissions biases are negative (see Fig. 9. b without dotted marks), which is additionally linked to the low modelled BDT fraction in these areas. In the Chaco region, the negative contribution of BDT highlights a possible underestimation of tree cover, which is evident when compared to TRIFFID and the ESA land-cover based PFT (see Fig. 1). In 2019, the tree cover fraction in Chaco was over 30%; however, Scrublands dominated in the Dry Chaco (38.2%) (San Martín et al., 2023). On average, for the study period, the TRIFFID modelled C4G, C4Pa, and bare soil as dominant PFTs with 88% cover in the Chaco region; none of the tree PFTs had an average fraction over 2%. Furthermore, since this region has emerged as a hotspot of deforestation and agricultural expansion (Baumann et al., 2017), in addition to more accurate PFTs, incorporating landscape fragmentation would also be potentially beneficial.

Despite the low correlation between soil moisture feature and the bias (R = 0.18), soil moisture is the second most significant feature in explaining these spatiotemporal biases in SA, particularly for North-SA and Mid-SA. For North-SA, the soil moisture



also presents a high but negative correlation with SHAP values (R = -0.70), indicating that the overestimations (underestimations) of CO emissions by INFERNO are related to drier (wetter) conditions. This suggests that fire emissions in North-SA are highly sensitive to soil moisture, leading to an inflated response. This aligns with the overestimations observed during the El Niño event in 2016, discussed in Section 3.1.2. In Mid-SA and South-SA, the correlation between SHAP values and soil moisture is low. This indicates that both dry and wet conditions contribute to negative and positive biases, suggesting a complex interaction with other variables included in the analysis. In South-SA, both low and high soil moisture particularly contribute to explaining positive biases (see Fig. 9. a, positive SHAP values quantiles). This complexity might be related to the role of precipitation/soil moisture in flammability and GPP, which have been observed to have different responses from different PFTs in the region (San Martín et al., 2023). Evidence of deficient representation of the complex interaction between soil moisture and fire in this region is the misrepresentation of the seasonal cycle (see Section 3.1.1 and Fig. S3), where flammability and GPP follow the precipitation peak.

The crop fraction, C4Cr, ranks third in contributing to the explanation of the CO emission bias from INFERNO in North-SA and South-SA; however, the contribution and the variable magnitudes were not significantly correlated, either positively or negatively. A few patches where C4Cr contributes the most are visible in Fig. 9.b, similarly for Sh in the south of the Arc of Deforestation. Since anthropogenic interactions are not associated with the simulated C4Cr, reducing bias through this variable would mean describing crop activities as harvesting (Li et al., 2013); and socioeconomic factors (Li et al., 2013). Furthermore, agricultural expansion and landscape fragmentation.

Although the INFERNO run did not include HDI, this feature appears to be around fourth place in terms of contribution for North-SA and Mid-SA, where this has demonstrated MB% reduction (see Section 2.5). HDI is then a prospective feature to address INFERNO biases.

## 4 Conclusions

We evaluated the fire CO emissions and sensitivity of the global fire model INFERNO estimations in South America (SA). The study quantified and assessed the spatiotemporal, seasonal cycle, and trend accuracy of the model's estimated CO emissions against five biomass burning inventories. For this, SA was divided into three subregions: North-SA, Mid-SA, and South-SA to compare differences in fire activity and biomes. With the least forest cover, South-SA exhibited the lowest disagreement in CO emissions between inventories, including FINNvn2.5 (Relative percentage range = 65%). The agreement was similar for North-SA and Mid-SA, if excluding FINNvn2.5. INFERNO was able to reproduce emissions in key active fire zones, such as deforestation fronts (e.g. Arc of Deforestation) and ecoregions like the Cerrado and Llanos, but likely underestimates CO emissions in the Chaco region, although still within GFEDvn4s range. Overestimation outside these regions, such as within the Amazon forest, led to enhanced CO emission overestimations, particularly in Mid-SA and North-SA. In Mid-SA, INFERNO demonstrated good performance reproducing the seasonal cycle of emissions, although with general overestimation of the magnitudes. In contrast, over North-SA, INFERNO exhibited a large spatiotemporal bias due to an erroneous bimodal representation of the seasonal cycle, while biases on South-SA were low despite the incorrect seasonal cycle. In both places,





the simulated CO emission closely follows both the flammability and GPP cycles; however, it was when GPP was high that the emissions incorrectly peaked.

INFERNO was able to reproduce the overall trend direction of CO emissions, although it erroneously reproduced an increasing trend near the Arc of Deforestation from 2004 to 2021. During the period from 2014 to 2021, INFERNO correctly estimated an increase in CO emissions along the SA low-level jet region. This region, which crosses Mid-SA and South-SA, has been particularly dry and vulnerable in recent years due to multiple meteorological factors, including the La Niña phase of the ENSO, as well as policy and socioeconomic factors. Due to the complexity of the fire regime in this region, INFERNO underestimates the magnitude of the trend in CO emissions but accurately identifies the direction of the trend. Over the short-term period, the inventories and satellite retrievals of TCOO disagree on the CO trend in North-SA; however, neither presents a significant result.

Multiple sensitivity experiments were conducted by modifying factors related to ignition, flammability, PFT, and also the individual factors: combustion completeness, average Burned Area, and emission factor. We evaluated the proposed use of the Human Development Index (HDI) in INFERNO, which improved performance in the Mid-SA and North-SA by reducing CO emissions; however, further reductions in CO emissions over South-SA resulted in poorer model performance. Additionally, the reduction in absolute bias observed when using constant anthropogenic and natural ignition was similar for all regions, although they described significantly different trends from the results with HDI in Mid-SA. Furthermore, the climatic datasets used for the control run, ERA5, demonstrate strong spatiotemporal performance. In contrast, using the W5E5 dataset to calculate flammability shows a lower bias in CO trends for Mid-SA and South-SA. The seasonal cycle across the three subregions was consistent for all climate input datasets, with ERA5 resulting in fewer monthly emissions. Changes in flammability were the most important factor driving changes in the simulated fire CO emissions seasonal cycle (MB%=116) compared with the experimental run. The experiment, which examines the effects of constant and varying factors on PFT, highlights the importance of forest cover (Broadleaf deciduous trees and Broadleaf - BDT evergreen tropical trees - BET-Tr) in determining the simulated fire CO emission magnitudes in SA. Using a prescribed PFT based on the satellite-based ESA Land Cover product, results in the highest spatiotemporal (MB%=115) and trend (MB%=167) changes against the control run, which are related to the relatively higher fraction of BET-Tr in Mid-SA and BDT in South-SA.

In line with the findings from the sensitive experiments, the feature importance analysis of the ML model indicated that BDT was the most significant feature contributing to the bias in INFERNO's CO emissions. A large (short) fraction of BDT contributes to overestimations (underestimations) of the emissions in SA. Both improving PFTs accuracy and incorporating the representation of human land-use management of the vegetation through variables, such as land fragmentation, might help reduce biases. Soil moisture was the second most significant contributor. In North-SA, the positive bias of CO emissions correlates with dry conditions, suggesting hypersensitivity to soil moisture. In South-SA, INFERNO biases exhibited a more complex relationship with soil moisture, which is likely associated with varying contributions of soil moisture/precipitation to GPP and flammability. C4 Crop contribution to the ML model emphasised the potential of including crop fire dynamics into the model to reduce biases. Similarly, the contribution of HDI suggests potential bias mitigation in North-SA and Mid-SA, as was observed in the sensitivity experiments.



This study highlights the capabilities and limitations of INFERNO in supporting the UKESM's new developments in a challenging region. Here, we conducted sensitivity experiments for various parameters and recommend a perturbed parameter ensemble method for a more in-depth evaluation of INFERNO's performance and uncertainty. Future research should also focus on accurately representing the seasonal cycle of fire activity in SA, addressing issues related to the role of precipitation on both GPP and flammability through soil moisture.

*Code and data availability.* The cut JULES-ES control configuration (based on JULES version 7.5) is stored at https://code.metoffice.gov. uk/trac/roses-u/browser/d/l/3/2/3/trunk (last access:11 March 2025). The fire CO emission are download from the inventories GFEDvn5 at https://surfdrive.surf.nl/files/index.php/s/VPMEYinPeHtWVxn, GFEDvn4s https://daac.ornl.gov/VEGETATION/guides/fire_emissions_v4_ R1.html (van der Werf et al., 2017), GFASvn1.2 https://rda.ucar.edu/datasets/d312009/dataaccess/ (Kaiser et al., 2012), FINNvn2.5 https: //ads.atmosphere.copernicus.eu/datasets/cams-global-fire-emissions-gfas?tab=overview (Wiedinmyer and Emmons, 2022). The 3BEM-FRP dataset was provided directly by the authors. The TCCO were downloaded from MOPPIT at https://asdc.larc.nasa.gov/project/MOPITT/ MOP02J_9. The data from the IASI retrieval product is now available at https://dx.doi.org/10.5285/4b31d47716604b9f84714fab39ce973c (Moore and Remedios, 2025). The HDI datasets were downloaded on a national scale https://datadryad.org/dataset/doi:10.5061/dryad.dk1j0 and on a subnational scale from https://globaldatalab.org/shdi/. Some assessments were done using the deforestation front for 2020 provided at and the ecoregion provided at https://globil.panda.org/datasets/panda::deforestation-fronts-2020-1/about. The model inputs are provided by ISIMIP3a at https://protocol.isimip.org/#/ISIMIP3a/fire

*Author contributions.* All the authors participated in reviewing and editing this manuscript. MPV: conceptualisation; data curation; formal analysis; investigation; methodology; software; visualisation; writing (original draft). RJP: conceptualisation; investigation; methodology; supervision; project administration. STT: conceptualisation; investigation; methodology; supervision; project administration. CD: methodology. DPM: resources. GM: resources. MPC: project administration

*Competing interests.* The contact author has declared that none of the authors has any competing interests

*Acknowledgements.* We thank the NCAS Computational Modelling Services Helpdesk for the software support regarding JULES-INFERNO. The authors also thank Camilla Mathison, Eleanor Burke, and Rich Ellis for their help in setting up the suite. This work was funded by the UK Natural Environment Research Council (NERC), which provided funding for the National Centre for Earth Observation (NCEO; grant no. NE/R016518/1, NE/X019071/1 and NE/R016518/1) and the NERC Panorama Doctoral Training Partnership (DTP; grant no. NE/S007458/1). The contributions of Steven Turnock were funded by the Met Office Climate Science for Service Partnership (CSSP) China project under the International Science Partnerships Fund (ISPF). São Paulo Research Foundation (FAPESP; grants 2019/25701-8, 2023/03206-0) funded the contributions from Guilherme Mataveli.





EUMETSAT provided spectral (L1C) and retrieved meteorological data (L2) for MetOp-B IASI. The IASI retrievals and JULES-INFERNO
runs were produced using JASMIN, the UK collaborative data analysis environment ( https://www.jasmin.ac.uk ), with the University of Le-
icester IASI retrieval Scheme (ULIRS). Pre-processing of IASI data into a structure suitable for ULIRS was performed using the ALICE
High Performance Computing facility at the University of Leicester.



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
