# Peer review of "Understanding drivers and biases of simulated CO emissions by the INFERNO fire model over South America"

_EGUsphere, 2025_

## Author Comment (AC1)

**Response to Reviewer #1**

We thank Reviewer #1 for their commitment to providing detailed comments and specific suggestions to improve this manuscript, with a focus on enhancing its readability. We have reproduced their comments below in black text and numbered Reviewer #1's comments for clarification when addressing comments relevant to both referees. Our responses are in blue text, and any additions to the manuscript are in red text. Our reference to line numbers is based on the initially submitted manuscript.

**General comment**

1)  Raw total column CO is not among the appropriate benchmarks for fire emissions. […].

We appreciate the concerns of Reviewer #1 and Reviewer #2 regarding the use of total column CO (TCCO) in our analysis. It is important to clarify that we include the TCCO not as a benchmark for evaluating emissions. The emissions and TCCO are linked as CO is emitted and then undergoes meteorological and chemical processing. So, while not directly comparable, the TCCO can provide useful information on emissions from space, increasing the robustness of the results. For instance, the TCCO trends for South America, where fire emissions dominate the CO balance (Lichtig et al, 2024), show general agreement with inventories and support a weak decrease trend in CO emissions. However, we recognise that our original wording of the introduction on line 68, where we stated, "we compare the carbon monoxide (CO) emissions from fires simulated by JULES-INFERNO with various biomass burning inventories and satellite-retrieved total column CO (TCCO)," was misleading. Therefore, while we believe the TCCO does have merit in this study, as both reviewers have questioned its application here, we have removed it from our analysis to avoid any confusion.

2)  There are multiple issues related to statistics
a.  Shapley additive explanation (SHAP) values are described as "the expected marginal contribution of a feature… It is calculated by taking the weighted average of all possible subsets of the selected features in which the specific feature can contribute." (l. 293-295) A SHAP value is not the expected marginal contribution of a feature. That is fortunate, because comparisons of marginal contributions among strongly correlated features easily becomes misleading. [⋯]. A package citation would be welcome.

Thanks for the comment; we recognised that we should use more precise language when describing this method to avoid incorrect explanations. We have reworded the sentence as:

These values are calculated as a weighted average of the differences in model predictions when the feature is included and excluded for all combinations of the remaining features.

" We employed the Shapley additive explanations (SHAP) method using the Python package from Lundberg et al. (2020). […]. SHAP is based on cooperative game theory and measures each feature's contribution to each prediction by calculating SHAP values. These values are calculated using a weighted average of the differences in predictions when the feature is added to all possible subsets of the remaining features (Lundberg et al. ,2020). We have used SHAP values to explain the dominant drivers in fire emissions in a consistent way with other recent studies, which also exploit machine learning methods for wildfire result applications (Wang et al., 2023, 2022; Liu et al., 2024)."

SHAP values can be calculated for a model whether or not the commendable complication of cross-validation is provided. The following asserted connection therefore does not hold. "We

calculate the SHAP values using a five-fold cross-validation approach... This allows us to calculate a complete set of SHAP values for the whole dataset " (l. 296 – 299)

We have also corrected the sentence that implied that cross-validation was required to calculate SHAP values. Below the new sentence added to l.296:

"SHAP values are computed for every prediction in the test set for every iteration of the five-fold cross-validation process."

b. Target Average Relative Ranges: Predictions that have perfectly accurate responsiveness to predictors reproduce only the portion of variability ($r_2$ ) that the predicting model can explain (Farmer & Vogel, 2016). [...]. If one wants realistic reactivity to predictors, it is a failure, not a success, if INFERNO's average relative range is as large of those of the emission inventories.

We agree with the well-explained comment; however, since we did not use the Average Relative Range (ARR) to compare the amplitude of the inventory variations with the model, this does not imply any change. Instead, the ARR was used to describe the average spread of the calculated annual emissions across the inventories in a specific area, as noted in line 238: "We use the Average Relative Range (ARR) of estimated fire CO emissions across different inventories to quantify the level of variation among them".

We think the confusion might be related to the explanation itself and the structure in which this is presented in the methodology. So, we tried to improve it by applying the suggestion to l. 238 presented in the section of the revision "line-specific comments" and changing part of the explanation, in addition to adding the following paragraph (where ARR abbreviation is replaced with Range% following Reviewer #1 4.c comment):

"Range% is designed to inform about the spread of estimated CO emissions magnitudes, considering the wide range of results from satellite-based estimations. The INFERNO simulations do not contribute to this metric."

    c. Feature exclusion: Your use of machine learning to analyse prediction residuals (Lundberg et al., 2020) lets you differentiate between the importance of omitted features and that of improvements that in theory may be possible via different handling of existing inputs. While extending your analysis may not be feasible, I wonder what might be learned by further exploration of the features with the highest SHAP scores.

Thank you for your suggestion. While the XGBoost approach combined with SHAP values is effective in highlighting the top features contributing to biases, this analysis is not isolated from the rest of the study. For example, sensitivity analysis also plays a key role in understanding how variables influence INFERNO. Rather than extending the SHAP-based exploration further, we have focused on synthesising these findings and providing a more comprehensive interpretation in the new discussion section.

An example illustrates that dropping an important but highly cross-correlated feature can lead to conclusions of dubious usefulness. The feature selection in this paper assumes that soil moisture represents the importance of leaf carbon as a diagnostic predictor in INFERNO. Within an ESM, soil moisture and carbon stock values have very different derivations. [...]. I suggest trying runs of XGBoost with all features, and/or analyzing results of the current runs with considerable attention to the implications of initial paring

We would like to thank reviewers #1 and #2 for their suggestions to run the XGBoost model using all variables without removing any correlated features. After further review of the literature, we found that gradient boosting models are well-suited for handling correlated features(Power et al., 2024). Based on the reviewers' comments, we have now used all INFERNO input variables within an XGBoost model, increasing the number of selected features used from 14 in the original analysis to 20. The results were compatible between the two XGBoost models using different numbers of inputs. The variables that ranked highest in the original feature importance analysis (based on SHAP values) retain their relative positions when using the full 20-input variables approach, as Fig. R1 illustrates. This approach also highlights the importance of other variables, such as Tropical Broadleaf Evergreen Trees. In the original manuscript, the contribution of this tree PFT was difficult to distinguish from its highly correlated soil moisture. Variables that also correlate with soil moisture, such as wood carbon, did not rank as high, which is another piece of new information that the new approach brings. Additionally, the model performs better, increasing the proportion of variability explained from around 64% to 67% based on the coefficient of determination $R^2$.

[Figure]

**Figure R1.** feature contribution to the XGBoost predicted biases using absolute SHAP values from the (a) original manuscript and (b) updated manuscript. (b) was updated in Fig. 9 of the original manuscript, Fig. 8 in the updated manuscript.

We add the following to the methodology section on l279:

"We selected 20 inputs for the machine learning model, comprising prescribed data and JULES outputs used by INFERNO to calculate emissions. [...]. We enable correlated features in the machine learning model, as in a gradient boosting model, any redundant information is automatically disregarded. This happens because the decision trees are built by splitting features in a series of dependent trees, so they can not make identical splits using correlated features (Power et al., 2024)."

Less critical concern: The choice of a VIF cutoff is a matter of professional opinion rather than a consequence of an objective discontinuity. A citation therefore is needed on l. 284. Your description is slightly muddled (l. 281-2). [...]

Since we are now using all features in the XGBoost model, this part has been removed from the manuscript.

d. Trend confidence intervals (might need only clarification): This paper compares trends found in the emissions inventories to trends in INFERNO's predictions (section 3.1.2). Fig. 4 shows that you have appropriately calculated trends one cell at a time (e.g., Andela et al. (2017)), which avoids pseudoreplication. There are two ways to calculate trend confidence intervals for large areas. I do not see the needed documentation of which you used. The conservative and prudent approach is to sum each year's emissions for the large region, then calculate a trend and its confidence interval by treating each year as one observation. [...]. Please address my uncertainty whether sound confidence intervals support, for example, the precision - and implication of comfortable fidelity - in l. 409-410's assertion that "most inventories and INFERNO suggested a negative trend of approximately -1.0%yr−1".

As inferred by Reviewer #1, we used an alternative method to calculate the trend in the emissions presented in Table 3. We used the "conservative and prudent approach" that Reviewer #1 mentioned: we summed the CO emissions for each year and region, then calculated the trend. We agreed that this information could have been clearer in the manuscript. This has been updated for the paragraph starting on l 246 to the following:

"To calculate the trends, we use the ordinary least squares linear regression (Perktold et al., 2024). This returns absolute trend values (Gg yr$^{-2}$), which we express as a percentage relative to the intercept (i.e., representing CO emissions at the start of the assessment period). We quantified the statistical significance of the CO emissions trend derived from both the inventories and INFERNO using the Mann-Kendall test at the 95% confidence level (Hussain and Mahmud, 2019). The trend and its significance were separately calculated for the individual grid point data and the regional cumulative emissions based on annual CO emissions."

Additionally, we have added the following note to Table 3 describing the information and the test used to calculate the statistical significance of the trends :

"Note: The magnitudes of trends are highlighted in bold when the Mann-Kendall test indicates a significant trend at the 95% confidence level (p-value < 0.05). Regionally aggregated annual CO emission time-series are used for the temporal trend analysis."

You calculate trends for two time periods, one only 8 years long. High interannual variability in fire incidence is globally generic, especially in forests. A result is that for trends annual fire incidence has a low signal to noise ratios. It is improbable that any real and persistent trend would become evident with statistical confidence in only 8 years.

We agree with Reviewers #1 and #2 that the 8-year period presents significant uncertainty due to the short time period and the high interannual variability in wildfire emissions.  Consequently, we have decided to exclude the short-term trend from our study.

e. Cross-inventory averaging (less critical concern): In places you average across four inventories and used the averages as the benchmark (e.g. l. 484). Where this is done, and why, needs to be indicated more clearly. [...]. I question including GFED4s in averages. Averaging implies that you have equal confidence in the accuracy of each included inventory. The description of GFED5 (Chen et al., 2023) dwells on the reasons for each change in the newer version. If you nevertheless feel GFED4s is worthy of inclusion in averages, that deserves justification. [...] Does including FINN enhance this study? Many papers already agree with you that FINN is out of line with other emission inventories (l. 483).[...]

In response to Reviewer #1's suggestion, which is in line with Reviewer #2's comments, we have removed the analysis on the FINN inventory from the main manuscript. Regarding the averaging across multiple fire emission inventories, we acknowledge that this is not an optimal benchmarking method, especially considering the methodological and regional differences among these inventories. However, in this study, we wanted to encapsulate the variability in estimates of CO emissions from fires across a range of the most up-to-date emission inventories. Both GFED4s and GFAS are well-established inventories in the literature, and while they have outperformed other inventories in emission evaluations in South America (Hua et al., 2023; Reddington et al, 2019). Their biases have also been highlighted (Naus et al., 2022, Liu et al., 2020). In contrast, 3BEM-FRP and GFEDvn5 represent newer-generation inventories adjusted with finer-resolution satellite data and updated land cover inputs. Because they include smaller fires, they tend to estimate higher emissions and are expected to be more accurate than GFEDvn4s and GFEDvn5; however, they lack the extensive and long-term validation that GFED4s and GFAS have. In fact, GFEDvn5 emissions are still under development and were downloaded as Beta products.

Therefore, we have preserved the individual characteristics of each inventory throughout the study, equally weighting GFEDvn4s, GFASvn1.2, 3BEM-FRP, and GFEDvn5 Beta in our averaged dataset used in the machine learning section, to represent a best estimate of fire CO emissions. This decision reflects a cautious and inclusive approach, acknowledging both the reliability of well-established inventories and the potential improvements offered by newer ones. Ideally, we would implement a performance-based weighting or subsampling strategy using an atmospheric model and compare the results with Total Columns CO (TCCO). However, this level of analysis is beyond the scope of our study and is therefore acknowledged in the limitations at the end of the manuscript.

Here is the new addition to l133 in the original manuscript:

"For the machine learning analysis of INFERNO biases (Section 2.6) and for visualising differences in selected figures, we calculated an ensemble average (mean) dataset based on the four inventories: GFED4s, GFED5, GFAS, and 3BEM-FRP. Each inventory was equally weighted in the average. GFED4s and GFAS are well-established inventories in the literature. While they have outperformed other inventories in their emissions estimate in South America (Hua et al., 2024; Reddington et al., 2019), their biases have also been noted (Naus et al., 2022; Liu et al., 2020). In contrast, 3BEM-FRP and GFED5 (the beta version) are considered next-generation inventories. They have been adjusted to better represent small fires and include updated and more accurate land cover data (Mataveli et al., 2023). However, these newer inventories lack the extensive long-term validation that GFED4s and GFAS have undergone. Overall, using an average of these inventories represents a balance between incorporating innovative methodologies and relying on well-established datasets for this study."

f. Geographic cutpoints (less critical concern, l. 88): Why geographic blocks rather than (groups of) PFTs, or the ecotypes shown in Figure S1? Why 3 instead of 2 or 5 regions? Replicating the division used in Li et al. (2024) (l. 90) might be useful if comparison to that work's findings were central to the analysis of this study's findings, but instead the reference is not subsequently mentioned. [...]

The geographic regions defined in our study are the same as those used in the studies of Li et al. (2024) and van Marle et al. (2017). While the scientific objectives of these studies might differ from ours, we feel that this is still a useful way to separate South America into distinct regions

for the scientific investigation of environmental parameters/variables. As Reviewer #1 points out, the separation for North-South America and Mid-South America allows for an important assessment of differences in the CO emission seasonal cycles. Additionally, the key benefit of using the definitions as in Li et al., (2024) and van Marle et al., (2017), is isolating the Arc of Deforestation zone in Mid-SA. Figure S1 (in the supplement of the original manuscript) shows the different fire-prone regions, and Reviewer #2 correctly points out that the Cerrado is split into two. However, the Mid-South America used here encompasses the most fire-prone zones of the Cerrado (Kim et al., 2025).

Still, we acknowledge that we need to better justify our choice of regions in the manuscript, which is why we have changed Section 2.1 l87 to the following:

"This study assesses the spatial distribution, seasonality and temporal evolution of CO fire emissions in continental South America (a key global region for fire activity and emissions). We focus on three regions: Northern South America (North-SA), Central South America (Mid-SA), and Southern South America (South-SA), as shown in Fig. 1. These regions are designed to capture emissions from the main fire-active regions of South America, while also accounting for unique fire patterns. These regions are consistent with previous work that evaluated the ability of CMIP models' to simulate fire emissions, including assessing trends and biases, across South America (Li et al., 2024; van Marle et al., 2017). The North-SA region experiences a unique fire season because its cycle is opposite to that of the southern region, due to the migration of the Intertropical Convergence Zone. The Mid-SA region encompasses fire emissions from the important Arc of Deforestation front (Pereira et al., 2022), an area threatened by continued land-use conversion and recognised as the world's largest savanna-forest transition (Marques et al., 2020). Importantly, defining Mid-SA as a broad but bounded region provides a practical scale for evaluating fire models like INFERNO, which are not designed for fine-scale simulations in highly variable zones such as the Arc of Deforestation. South-SA includes an important source of fire emissions from the Chaco biome. Although the division between Mid-SA and South-SA intersects the Cerrado fire-prone ecoregion, Mid-SA includes the deforestation front in the northern part of the Cerrado, where fires occur more frequently (Kim et al., 2025).

3) The paper provides too little assessment of the study's results in the context of other research. [...]. Does this work recommend any adjustment to the response shape for fuel load or soil moisture (Teckentrup et al., 2019, Fig. 5)? How do your findings support or differ from other ESM fire model sensitivity analyses? Overall, the analysis feels partly digested. Comparisons to previous work will help clarify and simplify what we have learned, and perhaps need to change, about INFERNO as a result of your efforts.

We thank Reviewer #1 for this valuable comment. We agree that better situating our findings within the context of previous research will strengthen the manuscript. In the revised version, we have added a dedicated section for discussion (updated Section 4) following Reviewer #2.1's suggestion. In this new section in l625, we compare our results to prior studies, including Teckentrup et al. (2019) and Forkel et al. (2019). Here is a section paragraph where comparisons with the literature can be observed:

"the XGBoost model identified soil moisture as the primary variable explaining the biases observed in INFERNO, while relative humidity also emerged as a significant factor related to meteorological conditions. In fact, Teckentrup et al. (2019) shows that increasing probability of fires driven by soil moisture begins to manifest in wetter conditions for INFERNO than for other models. The lack of global, long-term soil moisture datasets has prevented the assessment of

the direct effect of soil moisture on the performance of FireMIP models (Hantson et al., 2020). Nonetheless, other variables that describe drought conditions have been assessed. According to Forkel et al. (2019)'s evaluation, for INFERNO, maximum temperature contributes the most to the simulated burnt area, even more than for other models. However, wet conditions, represented by the number of days with significant precipitation, had little influence on burnt area (Forkel et al., 2019). Although reported with different variables, the role of dry conditions in INFERNO, identified by Forkel et al. (2019), supports the observed sensitivity to soil moisture, as this was also particularly evident in INFERNO simulations under dry conditions. Still, both studies differ in evaluation targets, the independent variables selected, and the variations in temporal and spatial resolution. Furthermore, burnt areas may be less sensitive to soil moisture than CO emissions, as evidenced by experiments that varied meteorological conditions. This observation is consistent with the results reported by Mathison et al. (2023), where INFERNO produced stable results for the burned area across various meteorological datasets."

4) More editing could make this paper a smoother reading experience

We have now gone through the manuscript and improved the readability of the text.

   a) Every figure would be improved by simplifying. If all the current detail really is needed as reference, move the details to the supplemental.

We have gone through the figures and simplified them where possible and reasonable.

      i) It is not apparent to me what you clear, simple intended take-away message is for any of the figures. For Figure 1, for example, what do you want a reader to notice about the contrast between the two maps – or is the message related to the distribution of PFTs and similarly apparent in each map?

Following the comment from Reviewer #1.4.a.iii, we have swapped Fig S1 with Fig 1.

      ii) To my thinking, Fig. S1 belongs in the main paper. Fig. 1 could move to the supplemental.

We agree and have swap Fig S1 with Fig 1.

      iii) For each figure that currently has more than one map, I suggest including only one.[...]

We respectfully disagree here. The current layout of figures with multiple maps have a clear focus of e.g. evaluation of the model, assessing variability between inventories or understanding the impacts of the sensitivity experiments. We have attempted to make the figures throughout the manuscript cleaner and simpler, but we feel that the multi-map panels provide use information and context in this work.

      iv) The black polygons in Fig. 5 are unreadable (and here and in Fig. 4 there are two sets of black polygons, one for nations). [...]

We use a cleaner hatching method in the figures (e.g. Figure 4) to indicate whether the results are statistically significant.

   b) Many paragraphs, especially in the results section, are simply difficult to follow. Others ramble and want tighter phrasing.

Following suggestions from both reviewers, we have updated the Results section according. Please see the "track changes" version of the updated manuscript to see our changes.

    c) The use of abbreviations is immoderate. The simplest illustration is to suggest that you compare the portion of sentences that contain at least one abbreviation from any page in this paper to a page of another paper you admire.

We have reduced the number of abbreviations in the document by incorporating the full labels for the following: aerosol optical depth (AOD), South America (SA), Machine Learning (ML), MATOPIBA and all the PFTs. Regarding the experiment names, although we did consider their meaning when creating them, we try to be more consistent this time, calling, for instance, IT-CST, BA-CST, and EF-CST the experiments that used a constant Ignition burnt area and emission factor. As we improve the experiment explanation in response to Reviewer #1's comment, the connection between the name and the experiment will be easier for the reader to identify.

    d) I regret that I continue to find it nearly impossible to follow what you are doing and why with each experiment. Below is one possible approach to reorganizing the portions of the methods section that describe INFERNO and the experiments.

We have made the objectives of each sensitivity experiment clearer in the updated manuscript. We have modified section "2.4 Sensitivity experiments on JULES-INFERNO" (l215). The experiments in this study are part of a sensitivity analysis for INFERNO using a one-at-a-time approach. The experiments are not intended to simulate a real-world scenario but to evaluate the model sensitivity by isolating the model response to the change of individual variables. Additionally, Table 2 (in the original manuscript) was modified for better clarity in summarising the experiments. We started the sections in l215 with the following two paragraphs:

"We conducted multiple experiments to assess the sensitivity of various processes and parameters controlling simulated fire emissions, and their roles in the INFERNO response. We did this using a one-at-a-time technique, varying individual parameters and variable inputs from the control simulation described in Section 2.3.1. The experiments are briefly summarised in Table 1. We evaluated the role of anthropogenic and natural ignition in different scenarios. First, we simulated total ignition with a constant global anthropogenic ignition rate of 1.5 ignitions/km$^2$/month, based on GFED estimates (Mangeon et al., 2016). For this, only IN varies in Equation 2, which we label IT-NAT. The role of both $I_A$ and $I_N$ was further evaluated by using a scenario with constant magnitudes for both ignitions. 1.5 ignitions/km2/month for $I_A$ and 2.7 flashes/km2/yr for $I_N$ . This experiment is called IT-CST. We also analyse socioeconomic scaling in ignition using the HDI as suggested by Teixeira et al. (2021). This experiment directly influences anthropogenic ignitions and fire suppression. The HDI was incorporated into the model using the dataset provided by Kummu et al. (2018). The experiment is labelled IT-HDI.

To analyse the role of the meteorological conditions and the uncertainty it can introduce into the simulations, we use different meteorological datasets from the ISIMIP3a climate-forcing dataset. In this, we compared the ERA5-based control with the simulations using W5E5 and 20CRv3. These experiments are named as F-W5E5 and F-20CR, respectively. Notice that, in addition to flammability in Equation 5, the meteorological conditions also affect ECP F T in Equation 6 through the carbon soil moisture and the available carbon."

    e) Many additional and salutary simplifications are readily available.
    i) Table 1: Is the NDT PFT applicable to this study? Any that are not can be omitted, with acknowledgement that the table is partial.

This table has been removed from the main manuscript. Both Needleleaf Tree PFTs are dominant in parts of the Andean region, as modelled by TRIFFID. Since we used all the PFT fractions in the XGBoost model, we found it beneficial to introduce them. However, for the new version of the manuscript, we have done it directly in the text in l168 as follows:

"For this setup, JULES-ES includes 13 plant functional types (PFTs), which include nine natural and four managed PFTs (Mathison et al., 2023). The natural PFTs are Broadleaf Deciduous Trees, Tropical Broadleaf Evergreen Trees, Temperate Broadleaf Evergreen Trees, Needleleaf Evergreen Trees, Needleleaf Deciduous Trees, Evergreen shrubs, Deciduous shrubs and C3 and C4 Grasses. The managed PFTs are C3 and C4 Crops and Pastures. C3 and C4 refer to photosynthetic pathways."

l. 153: "The JULES-ES configuration for ISIMIP3a was utilised in this study. This configuration is described in Mathison et al. (2023)…" to "We used the JULES-ES configuration for ISIMIP3a (Mathison et al., 2023)" (then maybe explain why).

 We have changed this accordingly in line with the reviewer's comment, see below:

"We used the JULES-ES configuration from the historical run of the third simulation round of the Inter-Sectoral Impact Model Intercomparison Project (ISIMIP3a) (Mathison et al., 2023), as it features a recent setup comparable to JULES in UKESM."

ii)  To make the inventory names easier to read, you might omit 'vn': GFED5 for GFEDvn5, and probably entirely drop the version references for GFAS and FINN after the first mention.

As suggested by Reviewer #1, we now use GFED4s, GFED5, and GFAS for the inventories instead of the longer labels of GFEDvn4s, GFEDvn5, and GFASvn1.2.

iii)  The numbers for RH_low and RH_high could replace the variable names, simplifying the Eq. 5.

These terms have been used in this form are based upon the studies that we have cited. Therefore, we believe that it is better to be consistent with previous work.

f)  Approaching the realm of picky details:

We thank Reviewer #1 for their attention to detail and have approved the manuscript accordingly. Please see the "track changes" version the updated manuscript.

g)  A thorough sweep would pick up many typos. There are also many opportunities to resolve awkward wording, especially choices of prepositions. […].

We thank the reviewer for providing a thorough proofreading of the manuscript. Below, we have checked (✔) the comment that has been corrected, and further descriptions of the change have been added to the comment where needed. Some other comments (X) have not been addressed, as the highlighted sentence has been deleted from the manuscript due to current changes.

We have addressed the typos and rewording suggested by Reviewer #1 where possible and reasonable to do so. We include suggestions to avoid the reader having to reread a sentence by removing parenthetical alternatives and joining numbers with their respective elements. We

have provided additional information on the location of the ecoregion mentioned and added Fig. S1 to the main manuscript. following the comment Reviewer #1 4.a.

- ✓ l. 5: "... South America (SA);, a region ..."
- ✓ l. 9: "most of the fire-active zones in ..." o l. 111: "While" appears to be an extra word.
- ✓ l. 124: "The EFs are consistently derived or partially derived from..." – wording is hard to follow

We have reworded this to:

"The EFs are commonly taken from Akagi et al. (2011) and Andreae and Merlet (2001)."

X    l. 139: Due to its wide swath, the global coverage is achieved in 12 hours

X    l. 144: Rather than decode all the abbreviations, which otherwise is needed, you might drop the description of all three products and simply say you chose the product that uses both thermal and near infrared bands.

X    l. 146: "...and an optimal estimation-based algorithm to retrieve..." "and optimization to retrieve"

- ✓ l. 173: "which models the PFTs competition"
- ✓ l. 240: missing 'd'
- ✓ l. 241: mismatch of singular pronoun with plural antecedent
- ✓ l. 244: 'Man-Kendall' has a typo

X    l. 314. "This" could refer to either of the two numbers in the previous sentence, the average relative range or the portion of total CO. The simplest way to prevent problems related to antecedents is to eschew generic pronouns.

X    l. 327: typo in the version number for FINN, and l. 498, typo in an experiment abbreviation - though I hope both have become moot.

- ✓ l. 439-440: "[I]nfractions against flora" looks like perhaps a literal translation. I do not know what it means and would prefer that the paper tell me the concept instead.

We reworded this sentence as:

"However, deforestation in Brazil escalated by approximately 80% in 2019, following the relaxation of forest protection regulations (Gatti et al., 2023)"

- ✓ l. 456: "Derived" needs rewording, because deforestation did not emit the air.

We reworded this sentence as:

"This was likely exacerbated by the reduction of warm and humid air transported from the Amazon, a situation significantly influenced by deforestation (Marengo et al., 2021) and the agricultural expansion in the region (Baumann et al., 2017)".

- ✓ l. 570: Hyperparameter tuning does not select model parameters. Do you mean "best hyperparameters"?

This sentence has been removed.

5) **Line-specific comments:**

a) l. 66: "This study decisively addresses this gap by rigorously assessing the performance of the model…" Isn't the qualitative judgment of 'decisively' for your readers to judge rather than you? The assertion sounds overstated, which almost cheapens your paper.

We can see the point the reviewer is trying to make but in this work we have undertaken a detailed assessment of the performance of INFERNO. However, we agree that "decisively" and "rigorously" are too strong a pair of terms. Therefore, these words have been removed from the sentence.

b) l. 150: "Fair" is generically a judgment call in the eyes of the beholder, and therefore best used very sparingly in academic papers. Here instead be more tangible about what is gained by your choice.

This text has been removed with the TCCO analysis.

c) near l. 183, Table 1: Why do some PFTs have emission factors of 0? Do those PFTs really need to be in the table at all? Before presenting Table 1, please tell the reader why we are seeing the information. For example, will you be changing each default value in an experiment? This suggestion is subordinate to restructuring the experiments introduction as recommended above.

Some of the sensitivity experiments performed using INFERNO resulted in changes to the factors listed in Table 1; however, we opted to include this table in the supplementary information to avoid complicating the main text. Instead, we incorporated the PFT names that were used within the paragraph (see Reviewer #1.4.e.i). Regarding the $EF_{CO}$ value of 0, we ran the control model and experiments using a configuration set of factors. For this, $EF_{CO}$ from Crops was set to 0 to facilitate comparisons with previous versions of JULES that did not include crops. We conducted an additional sensitivity experiment to assess the impact of the omitted fire CO emissions from crops. Results from this experiment indicated that the impact of including fire emissions of CO from crops was insignificant, as the main sources of CO emissions are the most forested areas. Details of this additional experiment were added to the supplementary information.

We acknowledge that we did not emphasise this point clearly, particularly since crop emissions were excluded from the results. Below is a new paragraph highlighting the exclusion of fire CO emissions from crops in this study.

"Table S1 presents the BAP F T and EF for each PFT modelled. The EFCO from C3-Crop and C4-Crop were not included in this model setup of JULES, which did not account for the crop PFTs. However, we conducted an experiment to assess the impact of fire CO emissions from crops. The experiment showed that excluding crops from simulations produced a negligible change in CO emissions across South America, increasing simulated CO emissions by only 1.4% for South-SA when included (see Fig. S1). However, it is important to note that INFERNO does not model crops differently from other PFTs; meaning that harvesting periods and crop seasonality are not included or represented. In this study, we are, however, evaluating the contributions of the crops' PFT fractions to the model biases (see Section 2.6)."

d) l. 234: Soil underlies virtually all land, so specify throughout that the PFT is 'bare' soil

We agree and have l. 234 label from soil to bare soil.

e) l. 238: Introduce each descriptive statistic first in terms of what you are trying to describe in the real world. In this instance, average relative range describes something like how well the model predicts the variability of a cell's emissions.

We thank Reviewer #1 for the suggestion. We think this way to introduce metrics makes it easier to understand the metrics themselves and their results. Below, we quote how we introduce the mean relative range. The abbreviation for this metric was changed to a simpler, more meaningful one (see Comment Reviewer #1.4.c).

"We use the mean relative range (Range%) to quantify the average variation in estimated annual CO emissions across inventories for a specified region or zone."

f) (l. 238 con't.) The description of average relative range's calculation seems to imply that the metric is a unified summary across the entire study area. You subsequently report relative ranges for particular ecotypes (e.g. l. 353). The calculations for areal subsets should be mentioned in the methods section.

We have made sure that it is clear that this metric is going to be used for multiple regions by describing the metric as quoted in Comment Reviewer #1.5.e.

g) l. 244, 246, 251: Please see first comment re l. 238. What aspect of fidelity to the real world (and not simply the model) do you want each metric to describe?

Following the suggestion of Reviewer #1.5.e we have introduced the remaining metrics as follows:

"We use CO emissions trends to assess the temporal evolution of CO emissions from inventories and INFERNO."

"We utilised the percentage mean bias (Bias%) to assess the model's representation of spatiotemporal variations, seasonal cycles, and trends of CO emissions "

"Similar to other studies (Hess et al., 2023; Liu et al., 2022), we utilised machine learning to identify the key factors causing annual biases in CO."

h) l. 285: Rather than a list of variables whose names are ambiguous out of context, describe the outcome of the step – for example, which variables you changed, or say you optimized all in the procedure's default list, or give some comparable description of why your choices matter.

In a similar way as suggested in Reviewer #1.5.e, we opted for not only listing but explaining what information every metric will provide to the model performance assessment. Below is how we now introduce the metrics:

"The model's performance was evaluated using the coefficient of determination ($R^2$), which indicates how much of the variability in the target CO emission biases is captured by the XGBoost model. Additionally, the Root Mean Square Error and Mean Absolute Error measure the average difference between the predicted and actual values, with the Root Mean Square Error being more sensitive to outliers."

i) l. 325: Fire frequency is a general term for a concept, but when associated with a specific number is spatially undefined. I think you mean fire return interval. The study you cite uses the

two terms appropriately. Also, Júnior et al. calculate frequencies for only a portion of Brazil's cerrado, and a portion whose fire management is not necessarily typical of the whole. It therefore seems inappropriate to describe the cited frequencies as describing to the whole ecoregion.

This has been removed.

j) l. 369: Most of INFERNO's emissions from South America are generated in the central region. [...]. You recount differences among the benchmark inventories in detail in this paragraph, but say nothing about why INFERNO is higher than all but (unreliable) FINN or column CO. Previously (l. 359) you propose that the source of inaccuracy may be "the model resolution and simplified process representation" – a degree of vague generality that is distressing. How can you use the seasonal cycle analysis to assess and/or refine this possibility? [...]. To which seasonable inputs is INFERNO excessively sensitive?

Following Reviewer #2.1, we have considerably reduced the comparisons between inventories to focus more on our aim target INFERNO, and as a response to Reviewer #1.2.e we have removed FINN from the study. We have changed. 359 to better express the implication of the model resolution, we wanted to emphasise:

" INFERNO generally captures the broad-scale features of emissions from this source, including its latitude-longitude range. This level of performance in relation to inventories is typical in global fire models like INFERNO. This is due to the inherent challenges in simulating a stochastic process, such as ignition, at a detailed scale using only vegetation data, meteorological information, and population density (Rovithakis et al., 2025). Therefore, simulating fine-scale features in a variable region, such as the Arc of Deforestation, are outside the scope of INFERNO."

Regarding the centre-South America seasonal cycle, the study still finds INFERNO results within the range of the inventory's variability. We understand the Reviewer's point that the model should not exceed observed response, but the inventories produce estimates that, although based on observations, are also calculated using supposition, parameterisation and models, so they are also full of uncertainty, and that is visible with the differences between the inventories themselves.  Below is the new paragraph in l383, presenting the seasonal cycle variability in the centre-South America:

"Figure 3 (d) highlights how INFERNO simulates a seasonal cycle peaking in September, similar to the inventories for Mid-SA. Within the inventory range, the GFED5 average seasonal cycle sits within the INFERNO interquartile range. In fact, between August and October, INFERNO and GFED5 show similar distributions (i.e., not significantly different). However, outside this period, INFERNO total monthly emissions are larger than all of the inventories. Spatially, Fig. 3 shows that INFERNO manages to simulate the timing of the CO emission peaks for the fire-active areas in Mid-SA. In fact, the absolute Bias% in the seasonal cycle amplitude is less than 10% compared to most of the inventories (see Table 2), except for comparison with GFED5. The GFED5 dataset exhibits a large seasonal cycle, particularly across the Arc of Deforestation, which contributes to higher average amplitudes."

However, based on the results we discussed (in the new discussion section l624) about the high sensitivity that INFENRO show to dry conditions, particularly represented by soil moisture:

"Our evaluation consistently shows that INFERNO-simulated CO emissions were too sensitive to drier conditions, particularly in Tree PFTs. In general, most overestimations relative to the inventories were in areas where modelled Broadleaf Deciduous Trees and Tropical Broadleaf Evergreen Trees dominate. The control of the Tree PFTs is particularly clear around the Arc of Deforestation in Mid-SA, where the relative fraction of these two tree PFTs appears to shape this main source of CO emissions in South America. In North-SA, with lower absolute emissions, the sensitivity of Tree-PFT to drought conditions underpinned the under-representation of the seasonal cycle. There, fire emissions from the Tree PFTs were incorrectly simulated with similar magnitudes to the fire-prone Llanos ecoregion emissions, where pasture is the dominant modelled PFT. Furthermore, the role of drought conditions in Tree PFTs was evident in the increasing biases over time, consistent with the growing influence of drought events. Finally, the large differences in simulated CO emissions resulting from different meteorological datasets used as inputs to JULES indicate that the response to drought conditions in Tree PFTs-dominated areas outweighs the influence of carbon availability, particularly in shaping the seasonal cycle of emissions. Compared with other models, the INFERNO fuel load index reaches its maximum much more rapidly in response to fuel density (Teckentrup et al., 2019), which is why, with high carbon availability, changes in fuel density may lead to muted differences."

k) l. 385: You seem to speculate that high gross primary productivity in September and October in the grasslands of northern South America drives the unrealistic late-season peak in INFERNO emissions. Those months are late in the rainy season. INFERNO lacks a mechanism to account for curing to recognize that grasses barely burn when green. Is that, together with the accumulation of grass growth over the course of the wet season, what you think causes the prediction error pattern?

We found two main error sources that affect the seasonal cycle simulated by INFERNO:

(1) high sensitivity to dry conditions, represented by soil moisture in Tree PFTs.

"For North-SA, the peak period of fire activity is represented, but with magnitudes higher than any of the inventories for January to April. Additionally, INFERNO simulates an erroneous second peak of emissions in October with similar magnitudes to the first peak. Although emissions in the Tree PFT domain to the east of the region tend to peak in October (Cummings et al., 2025) (see Fig. S3), these are negligible compared to the regional emissions around March.[...]" [for October,] "however, JULES simulates particularly dry conditions in eastern North-SA, resulting in relatively low Gross Primary Production (GPP). This means that dry conditions, rather than carbon availability, might be leading to the overestimations in this period."

The sensitivity experiments also support our finding, as the dryer reanalysis 20Crv3 (experiment F-20Cr), particularly enhanced the wrong peak of emissions (l525):

"Remarkably, in North-SA, the distinct meteorological conditions in F-20Cr resulted in higher magnitudes of CO emissions at the wrong annual peak of emissions, as Tree PFTs are particularly influenced by drier conditions."

We also use the literature to discuss this finding further in our new discussion section (l624):

"Our evaluation consistently supports that INFERNO-simulated CO emissions were too sensitive to drier conditions, particularly in Tree PFTs. In fact, Teckentrup et al. (2019) shows

that increasing probability of fires driven by soil moisture begins to manifest in wetter conditions for INFERNO than for other models."

(2) Sensitivity to carbon availability when the Tree PFTs fraction is low.

We discussed it on our new discussion section (l624):

"Our findings indicate that the simulated seasonal cycle of CO emissions by INFERNO closely follows the rise in GPP and flammability during December-January, a period marked by high temperatures and rainfall. As shown by Teckentrup et al. (2019), the INFERNO fuel load index, which is used in the flammability calculation, is highly sensitive to minor changes in fuel density when the overall fuel amount is small. Therefore, even during the rainy season, there may be hot days without rain, during which the relative humidity is sufficiently low to enable flammability to peak according to the fuel load index."

l) near l. 400, Figure 3: Please see note above about complexity of figures. Also, the horizontal lines for the means (?medians) are too hard to see. Please include the percentile cutpoints for the coloredportion of the boxes in the caption. Please make the panel letters bigger. I would find panel (a) easier to digest with a linear y-axis, with different y-axis ranges for each region.

We have changed this figure by removing the box plot panel and adding a map of the months with the highest CO emissions, based on the average of the inventories and INFERNO. The map provides the needed information to explain the incorrect seasonal cycle shapes in North-SA and South-SA.

m) near l. 435, caption for Table 3: Please note that the time period for mean emissions and seasonal amplitude is the full study period. Be explicit what the parentheses mean, which I assume to be standard errors of the mean, and bold, which I assume means statistical confidence that a value differs from zero. Why are standard errors for trends not included?

Thanks for identifying the missing information. Additionally, to add the confidence intervals in the table, we have rewritten the footnote as:

"Note: The magnitudes of trends are highlighted in bold when the Mann-Kendall test indicates a significant trend at the 95% confidence level (p-value < 0.05). Regionally aggregated annual CO emission time series are used for the temporal trend analysis. In the table, 'SD' and 'confidence' refer to the standard deviation values and the 95% confidence intervals."

n) (Table 3, con't) Here or somewhere else, I would like to see simple correlations of cell-month values between INFERNO and each inventory.

We followed Reviewer #1's suggestion to add useful evaluation information to this Table; however, instead of correlation, we used normalised mean error to compare the results with other studies, and we also reported the mean bias percentage. Please look at Table 2 in the new manuscript to see the additions.

o) (Table 3, con't) The time scales differ between Figure 3 and Table 3 – monthly in one, annual in the other. Unless there is a strong reason, make them consistent. After adjusting for time scales, is the data in the rows for emissions and seasonal amplitude in Table 3 the same as displayed in Figure 3? If not, I'm confused. If so, duplication in the

table is a distraction. If you feel strongly that the numbers need to be documented in tabular as well as graphical form, move them to the supplemental. After that and after removing data related to column CO, the remaining data in Table 3 might more effectively be presented graphically.

Both Figure 3 and Table 3 have changed; Table 3 still presents the amplitude of the seasonal cycle of CO emissions but additionally has some statistics that evaluate INFERNO results. Figure 3 presents the average monthly CO emission from where the amplitude can be calculated. However, we decided to keep the information in Table 3 to provide more complete information, including magnitudes and biases.

p) l. 448, Fig 5: Might these maps be more informative as differences from INFERNO? Maybe as a single map, INFERNO minus the mean of 4 inventories?

We have added a more informative difference map for this and Figure 2. However, we are retaining the results from each inventory and INFERNO as justified in the response to Reviewer #1 4.a comment.

q) l. 448: Where do we get a mean of 61% per year (or 85% on l. 451) when the map color scale tops out at 15 and much of the southern region has light colors? l. 452: Please give the baseline for the 208% change. Is it the 4-inventory mean?

This paragraph has been removed, following the advice of the reviewers Reviewer #1.2.d, because the trend was calculated from a shorter time period (8 years). As described in the response to the same comment, the regional trends were calculated based on the total annual CO emissions for the desired regions, while the figure presents pixel-by-pixel trends. We have emphasised more the different methods in the new versions of the manuscript l 246 as quoted in Reviewer #1.2.d response.

r) near l. 487, Fig. 6: I find the version in S6 plus a map of the baseline case to be more meaningful.

We have swapped Fig. 6 and Fig. S6 as recommended.

s) near l. 507, caption for figure 7: The word "type", like "class", "group", etc., conveys no inherent meaning. Please choose a more informative label. While this figure needs further simplification, currently it is the clearest of the figures. I suggest you muse about how to draw greater focus on the central region throughout the paper, to highlight its disproportionate contribution to continental totals and tendencies.

We have removed the "type" categorisation from the label. This is now described as:

"Figure 6. CO emissions seasonal cycle modelled by the different INFERNO experiments and control run in (a-d) North-SA, (e-h) Mid-SA and (i-l) South-SA."

t) l. 539: "This can cause a spatiotemporal overestimation of around 50%." I'm not sure what you mean in referring to a spatiotemporal overestimation. Is 50% the maximum for any cell-year, for instance?

This 50% overestimation was determined by comparing the biases from the inventory's estimations for both the control and NO-FDBK experiments. 50% was the maximum difference between them, particularly compared with GFAS, however the average difference was around 30%. We can see that it is not clear what we meant by that 50% overestimation so we reworded

this and also we decided better to present the average magnitude of the bias for Mid-SA to highlight the region with the largest emissions. Below is the new paragraph:

"Although all experiments lack feedback from the atmosphere, as this is a land-only model, the results from the NO-FDBK experiment illustrate the proportional error that the omission of feedback to land can introduce into the fire model [...] The No-FDBK experiment results in spatiotemporal CO emissions that are approximately 94% higher than the inventory estimates for Mid-SA, which is 22% higher than the Bias% observed in the control run (~72%) compared to the inventories (see Table 2)."

> u) l. 570: Hyperparameter tuning does not select model parameters. Do you mean "best hyperparameters"?

We meant hyperparameter, thanks for the correction. The model parameters are learnt in the training process. We have rewritten this as:

"We ran hyperparameter tuning to select the hyperparameters that lead to the best model performance."

> v) near l. 590, Fig 9: I strongly suggest no abbreviations in the legend(s). By rearranging, you may not need to show the legend labels twice.

Thanks for the suggestion. We included the full name of the feature in the figure, excluding only those that contribute less to explaining the biases. This uses the full-variable machine learning version suggested in Reviewer #1.2.c. We also rearranged to present the feature label once.

> w) l. 630-631: "INFERNO was able to reproduce emissions in key active fire zones, such as deforestation fronts (e.g. Arc of Deforestation)…" To your satisfaction, really?? More generally, this section summarizes your discoveries about which subsets of predictions tend most or least to be accurate. It says almost nothing about what the accuracy patterns mean. Why predictions are worse or better when and where they are, and ideally therefore how to improve the model, are more useful insights. I think you have a basis to make reasonable speculations.

Following Reviewer #2.1, we have split the Results from the Discussion section to bring out a deeper discussion of the findings across the study. In this section, we have presented our findings and discussed them in a more objective way. Additionally, the new section provides clear suggestions for addressing biases. Below, we quote the opening paragraph of the Discussion section, where we discuss the INFERNO simulated CO emissions spatiotemporal and seasonal performance around the Arc of Deforestation.

"The results of this study were consistent in showing that dry conditions, particularly soil moisture, and Tree PFTs in South America are the top drivers of biases in INFERNO. The few factors used by INFERNO that represent anthropogenic influence on fire ranked after those; however, including more socioeconomic factors could help explain the remaining 33% of the bias in the XGBoost model.[...]

Our evaluation consistently shows that INFERNO-simulated CO emissions were too sensitive to drier conditions, particularly in Tree PFTs. In general, most overestimations relative to the inventories were in areas where modelled Broadleaf Deciduous Trees and Tropical Broadleaf Evergreen Trees dominate. The control of the Tree PFTs is particularly clear around the Arc of Deforestation in Mid-SA, where the relative fraction of these two tree PFTs appears to shape this

main source of CO emissions in South America. In North-SA, with lower absolute emissions, the sensitivity of Tree-PFT to drought conditions enhanced the under-representation of the seasonal cycle. There, fire emissions from the Tree PFTs were incorrectly simulated with similar magnitudes to the fire-prone Llanos ecoregion emissions, where pasture is the dominant modelled PFT. Furthermore, the role of drought conditions in Tree PFTs was evident in the increasing biases over time, consistent with the growing influence of drought events. Finally, the large differences in simulated CO emissions resulting from different meteorological datasets used as inputs to JULES indicate that the response to drought conditions in Tree PFTs-dominated areas outweighs the influence of carbon availability, particularly in shaping the seasonal cycle of emissions. Compared with other models, the INFERNO fuel load index reaches its maximum much more rapidly in response to fuel density (Teckentrup et al., 2019), which is why, with high carbon availability, changes in fuel density may lead to muted differences"

**Added References**

Cummings, A. R., Kennady, B. J., and Adeuga, A. M.: Fire Regions of a Northern Amazonian Landscape Relative to Indigenous Peoples' Lands, Remote Sensing, 17, https://doi.org/10.3390/rs17193386, 2025.

Gallup, S. M., Ford, B., Naus, S., Gallup, J. L., and Pierce, J. R.: Equations to Predict Carbon Monoxide Emissions from Amazon Rainforest Fires, Fire, 7, https://doi.org/10.3390/fire7120477, 2024.

Forkel, M., Andela, N., Harrison, S. P., Lasslop, G., van Marle, M., Chuvieco, E., Dorigo, W., Forrest, M., Hantson, S., Heil, A., Li, F., Melton, J., Sitch, S., Yue, C., and Arneth, A.: Emergent relationships with respect to burned area in global satellite observations and fire-enabled vegetation models, Biogeosciences, 16, 57–76, https://doi.org/10.5194/bg-16-57-2019, 2019.

Haas, O., Prentice, I. C., and Harrison, S. P.: Global environmental controls on wildfire burnt area, size, and intensity, Environmental Research Letters, 17, 065 004, https://doi.org/10.1088/1748-9326/ac6a69, 2022.

Hussain, M. and Mahmud, I.: pyMannKendall: a python package for non parametric Mann Kendall family of trend tests., Journal of Open Source Software, 4, 1556, https://doi.org/10.21105/joss.01556, 2019.

Liu, Z., Zhou, K., Yao, Q., and Reszka, P.: An interpretable machine learning model for predicting forest fire danger based on Bayesian optimization, https://doi.org/10.48130/emst-0024-0026, 2024.

Lundberg, S. M., Erion, G., Chen, H., DeGrave, A., Prutkin, J. M., Nair, B., Katz, R., Himmelfarb, J., Bansal, N., and Lee, S.- I.: From local explanations to global understanding with explainable AI for trees, Nature Machine Intelligence, 2, 2522–5839, https://doi.org/10.1038/s42256-019-0138-9, 2020

Perktold, J., Seabold, S., Sheppard, K., ChadFulton, Shedden, K., jbrockmendel, j grana6, Quackenbush, P., Arel-Bundock, V., McKinney, W., Langmore, I., Baker, B., Gommers, R., yogabonito, s scherrer, Zhurko, Y., Brett, M., Giampieri, E., yl565, Millman, J., Hobson, P.,

Vincent, Roy, P., Augspurger, T., tvanzyl, alexbrc, Hartley, T., Perez, F., Tamiya, Y., and Halchenko, Y.: statsmodels/statsmodels: Release 0.14.2, https://doi.org/10.5281/zenodo.10984387, 2024.

Power, J., Côté, M.-P., and Duchesne, T.: A Flexible Hierarchical Insurance Claims Model with Gradient Boosting and Copulas, North American Actuarial Journal, 28, 772–800, https://doi.org/10.1080/10920277.2023.2279782, 2024.

Reddington, C. L., Morgan, W. T., Darbyshire, E., Brito, J., Coe, H., Artaxo, P., Scott, C. E., Marsham, J., and Spracklen, D. V.: Biomass burning aerosol over the Amazon: analysis of aircraft, surface and satellite observations using a global aerosol model, Atmospheric.

Rovithakis, A., Burke, E., Burton, C., Kasoar, M., Grillakis, M. G., Seiradakis, K. D., and Voulgarakis, A.: Estimating future wildfire burnt area over Greece using the JULES-INFERNO model, Natural Hazards and Earth System Sciences, 25, 3185–3200, https://doi.org/10.5194/nhess-25-3185-2025, 2025.

Teckentrup, L., Harrison, S. P., Hantson, S., Heil, A., Melton, J. R., Forrest, M., Li, F., Yue, C., Arneth, A., Hickler, T., Sitch, S., and Lasslop, G.: Response of simulated burned area to historical changes in environmental and anthropogenic factors: a comparison of seven fire models, Biogeosciences, 16, 3883–3910, https://doi.org/10.5194/bg-16-3883-2019, 2019."This was likely e

van Marle, M. J. E., Kloster, S., Magi, B. I., Marlon, J. R., Daniau, A.-L., Field, R. D., Arneth, A., Forrest, M., Hantson, S., Kehrwald, N. M., Knorr, W., Lasslop, G., Li, F., Mangeon, S., Yue, C., Kaiser, J. W., and van der Werf, G. R.: Historic global biomass burning emissions for CMIP6 (BB4CMIP) based on merging satellite observations with proxies and fire models (1750–2015), Geoscientific Model Development, 10, 3329–3357, https://doi.org/10.5194/gmd-10-3329-2017, 2017.

Wang, S., Foster, A., Lenz, E. A., Kessler, J. D., Stroeve, J. C., Anderson, L. O., Turetsky, M., Betts, R., Zou, S., Liu, W., Boos, W. R., and Hausfather, Z.: Mechanisms and Impacts of Earth System Tipping Elements, Reviews of Geophysics, 61, e2021RG000 757, https://doi.org/10.1029/2021RG000757, e2021RG000757 2021RG000757, 2023.

Wang, S. S.-C., Qian, Y., Leung, L. R., and Zhang, Y.: Interpreting machine learning prediction of fire emissions and comparison with FireMIP process-based models, Atmospheric Chemistry and Physics, 22, 3445–3468, https://doi.org/10.5194/acp-22-3445-2022, 2022.

---

## Author Comment (AC2)

**Response to Reviewer #2**

We thank Reviewer #2 for their valuable comments and suggestions, which have helped make the study clearer and more organised. We believe the manuscript has improved significantly after addressing these points, and we appreciate the opportunity to convey our message more effectively. We have reproduced Reviewer #2's comments below in black text and numbered the comments for clarification when addressing comments relevant to both referees. Our responses are in blue text, and any additions to the manuscript are in red text. Our reference to line numbers is based on the initially submitted manuscript.

**General comments**

1. The results and discussion section is incredibly dense and hard to follow. The authors have performed so many sensitivity tests and created so many acronyms for them that it is hard for the reader to understand what's going on and the aims of the paper risk being buried. Additionally, many of the plots are very busy (although I appreciate that large-format versions would be available in the final online version of the paper). At the very least, I would suggest the authors seriously consider whether all the information in the results section is necessary to be included in the main section of the paper and consider moving some plots to the supplementary info. I would suggest splitting the results and discussion, which would enable the thread of the paper to be easier to follow. The paper would also benefit from a clearer statement of the objectives and demonstrating how each analysis section is designed to address them.

We thank Reviewer #2 for their various suggestion to improve the readability, clarity and structure of this manuscript. We have made numerous changes to the manuscript following both reviewers' comments to make it easier to follow. Below, we highlight the major changes:

   i. We have reduced the number of sensitivity tests by eliminating some redundant experiments related to the factors BA, EF, and HDI. The sensitivity analysis section has also improved the explanation of the experiments following Reviewer #1 4.d comment and the result section, where we focus our analysis on the comparison between the experiments and the control run.
   ii. We focused more on the comparison between INFERNO and the inventories than on the comparison between the inventories, as this is not the main aim of the current study.
   iii. We have reduced the use of acronyms. We changed the abbreviation of some PFTs and experiments, removed the abbreviation from some PFTs and infrequently used words (more information in the response to Reviewer #1 4.c).
   iv. We have changed some plots, making them simpler to interpret (e.g. Fig. 3 and 8). We have also removed and added supplementary figures in the main manuscript, which better support the analysis (more information about changes to the plots can be found in the response to Reviewer #1 4.a).
   v. As suggested by Reviewer #2, we have separated the results and discussion sections for better clarity.

Finally, we have now clearly stated our objectives in the last paragraph to the introduction, outlining the sections intended to address them. Below, we show the change to the last paragraph of the introduction in the revised manuscript (l65):

"[...] previous studies have primarily focused on carbon emissions from fires, whereas this study aims to evaluate the simulation of fire-derived emissions in atmospheric models. We

seek to identify areas for development and improvement by analysing the biases associated with these emissions. In Section 3, we present our results focusing on the evaluation of INFERNO against fire emission inventories, use sensitivity experiments to investigate key drivers influencing CO emissions and finally quantify the model processes contributing to the calculated model-inventory biases. [...]"

2.  A large section of the paper is devoted to comparing different satellite-derived fire products against each other. While this is valuable, I think some of the finer details could be moved to a methodological note in the supplementary info. As is often the case, the authors find that the differences between the observational products are about the same as the differences between the model and the observations. With this in mind, the approach used which averages multiple fire datasets to produce a benchmark is not sufficiently justified. The authors exclude FINN (in which case, it could be removed from the analysis in the main text) but include GFED4 in the averaging even though it differs substantially from GFED5 (for example). The reasoning behind the averaging of multiple fire products needs to be more clearly explained.

We thank Reviewer #2 for pointing out the potential misalignment in our inventory comparison with our objective of evaluating INFERNO. Therefore, we have significantly reduced these comparisons in the manuscript, but still provide a small section on comparing inventories to acknowledge the spread in emission estimates in Section 3.1.

In response to Reviewer #1's suggestion, which is in line with Reviewer #2's comments, we have removed the analysis on the FINN inventory from the main manuscript. Regarding the averaging across multiple fire emission inventories, we acknowledge that this is not an optimal benchmarking method, especially considering the methodological and regional differences among these inventories. However, in this study, we wanted to encapsulate the variability in estimates of CO emissions from fires across a range of the most up-to-date emission inventories. Both GFED4s and GFAS are well-established inventories in the literature, and while they have outperformed other inventories in emission evaluations in South America (Hua et al., 2023, Reddington et al, 2019). Their biases have also been highlighted (Naus et al., 2022, Liu et al., 2020). In contrast, 3BEM-FRP and GFEDvn5 represent newer-generation inventories adjusted with finer-resolution satellite data and updated land cover inputs. Because they include smaller fires, they tend to estimate higher emissions and are expected to be more accurate than GFEDvn4s and GFEDvn5; however, they lack the extensive and long-term validation that GFED4s and GFAS have. In fact, GFEDvn5 emissions are still under development and were downloaded as Beta products.

Therefore, we have preserved the individual characteristics of each inventory throughout the study, equally weighting GFEDvn4s, GFASvn1.2, 3BEM-FRP, and GFEDvn5 Beta in our averaged dataset used in the machine learning section, to represent a best estimate of fire CO emissions. This decision reflects a cautious and inclusive approach, acknowledging both the reliability of well-established inventories and the potential improvements offered by newer ones. Ideally, we would implement a performance-based weighting or subsampling strategy using an atmospheric model and compare the results with Total Columns CO (TCCO). However, this level of analysis is beyond the scope of our study and is therefore acknowledged in the limitations at the end of the manuscript.

Here is the new addition to l133 in the original manuscript:

"For the machine learning analysis of INFERNO biases (Section 2.6) and for visualising differences in selected figures, we calculated an ensemble average (mean) dataset based on the four inventories: GFED4s, GFED5, GFAS, and 3BEM-FRP. Each inventory was equally weighted in the average. GFED4s and GFAS are well-established inventories in the literature. While they have outperformed other inventories in their emissions estimate in South America (Hua et al., 2024; Reddington et al., 2019), their biases have also been noted (Naus et al., 2022; Liu et al., 2020). In contrast, 3BEM-FRP and GFED5 (the beta version) are considered next-generation inventories. They have been adjusted to better represent small fires and include updated and more accurate land cover data (Mataveli et al., 2023). However, these newer inventories lack the extensive long-term validation that GFED4s and GFAS have undergone. Overall, using an average of these inventories represents a balance between incorporating innovative methodologies and relying on well-established datasets for this study."

3. It is unclear why total column CO observations are being used in this analysis. The modelling setup used produces CO emissions, but as it is a land-only model it is confusing to the reader to imply that atmospheric modelling has been done as well. To evaluate the model using TCCO you would need to simulate the atmosphere as well. Atmospheric transport of CO, a species with a long atmospheric lifetime, along with the topography and regional circulation of SA, means that TCCO is not necessarily co-located with peak CO emissions from fires. Additionally, some of the analysis is limited by the need to consider shorter timescales as a result of limitations of the TCCO datasets. I suggest the authors remove the TCCO analysis from the paper and stick to CO emissions. Using the emissions produced by INFERNO to force an atmospheric chemistry model, and comparing that to observed TCCO, would be a logical next step but would be beyond the scope of this paper.

We appreciate the concerns of Reviewer #1 and Reviewer #2 regarding the use of total column CO (TCCO) in our analysis. It is important to clarify that we include the TCCO not as a benchmark for evaluating emissions. The emissions and TCCO are linked as CO is emitted and then undergoes meteorological and chemical processing. So, while not directly comparable, the TCCO can provide useful information on emissions from space, increasing the robustness of the results. For instance, the TCCO trends for South America, where fire emissions dominate the CO balance (Lichtig et al, 2024), show general agreement with inventories and support a weak decrease trend in CO emissions. However, we recognise that our original wording of the introduction on line 68, where we stated, "we compare the carbon monoxide (CO) emissions from fires simulated by JULES-INFERNO with various biomass burning inventories and satellite-retrieved total column CO (TCCO)," was misleading. Therefore, while we believe the TCCO does have merit in this study, as both reviewers have questioned its application here, we have removed it from our analysis to avoid any confusion.

4. I will confess that I am not an expert in the use of ML detailed in this work. However, the way in which certain variables were excluded from the ML model seems to be limiting. My understanding of this approach is that it can handle large numbers of covariates and indeed that this is a strength of the approach. It would be interesting to see if the results are sensitive to the choice of variables excluded, or to run the analysis with all variables. The use of soil moisture to represent leaf and wood carbon in particular is a concern – these variables strongly covary, but are calculated in very different ways in the land

model. By using one set of processes to represent another, the authors may be limiting the power of the analysis especially given the goal is to evaluate model processes.

We would like to thank reviewers #1 and #2 for their suggestions to run the XGBoost model using all variables without removing any correlated features. After further review of the literature, we found that gradient boosting models are well-suited for handling correlated features(Power et al., 2024). Based on the reviewers' comments, we have now used all INFERNO input variables within an XGBoost model, increasing the number of selected features used from 14 in the original analysis to 20. The results were compatible between the two XGBoost models using different numbers of inputs. The variables that ranked highest in the original feature importance analysis (based on SHAP values) retain their relative positions when using the full 20-input variables approach, as Fig. R1 illustrates. This approach also highlights the importance of other variables, such as Tropical Broadleaf Evergreen Trees. In the original manuscript, the contribution of this tree PFT was difficult to distinguish from its highly correlated soil moisture. Variables that also correlate with soil moisture, such as wood carbon, did not rank as high, which is another piece of new information that the new approach brings. Additionally, the model performs better, increasing the proportion of variability explained from around 64% to 67% based on the coefficient of determination $R^2$.

[Figure]

**Figure R1.** feature contribution to the XGBoost predicted biases using absolute SHAP values from the (a) original manuscript and (b) updated manuscript. (b) was updated in Fig. 9 of the original manuscript, Fig. 8 in the updated manuscript.

We add the following to the methodology section on l279:

"We selected 20 inputs for the machine learning model, comprising prescribed data and JULES outputs used by INFERNO to calculate emissions. [...]. We enable correlated features in the machine learning model, as in a gradient boosting model, any redundant information is automatically disregarded. This happens because the decision trees are built by splitting features in a series of dependent trees, so they can not make identical splits using correlated features (Power et al., 2024)."

5. The paper contains a number of typos and instances of awkward wording, which are fixable with thorough proof-reading. I do not believe it is the responsibility of peer

reviewers to perform the work of a sub-editor, though I have pointed some examples below.

We have now gone through the manuscript and updated the text where appropriate.

**Specific comments**

We thank the reviewer for going further in their role by pointing out typos and sentences that required rewording and editing. Below, we have checked (✓) the comment that has been corrected. Further descriptions of the change have been added to the comment where needed:

- ✓ Some of the authors' names seem to be misspelled in the submission, based on the professional profiles found on the websites of their universities ('Veláquez-García' = Velásquez-García, 'Chiperffield' = Chipperfield).
- ✓ Line 7: Specifically, this should say 'carbon monoxide (CO) emissions estimates'.
- ✓ Line 10: Remove 'categories of'.
- ✓ Lines 16-18: It is not immediately clear what the percentages in this sentence refer to.

The percentage mentioned in the abstract refers to the mean bias percentage. Since this is a measure of bias or relative difference, we have chosen to refer to it directly as such. Below is an example of how the percentages are currently presented in the abstract:

"[...] in the Arc of Deforestation (southern Amazon), INFERNO tends to overestimate CO emissions by around 70%. [...]. An experiment shows that INFERNO produces up to 100% higher CO emissions in the Amazon region when using drier meteorological reanalyses"

and refer to the metric in the brackets. Note that instead of comparing the experiments and the control based on their performance (i.e. comparing with the inventories), we are now comparing the simulations directly, which is why these numbers have changed.

- ✓ Lines 26-27: Unclear wording ('the success of fire-prone ecosystems is enhanced').

We meant "succession"; however, we removed this sentence while rewriting a part of the manuscript to make the objective of this study clearer (Review #2.1) and passed the manuscript through proofreading.

- ✓ Line 45: Suggest 'represent' or 'simulate' instead of 'understand'.

We have chosen "represent" as a more accurate word for the message.

- ✓ Line 61: Where does the figure that SA represents 15% of annual fire carbon emissions come from? Over what time period?

Thanks for highlighting the missing reference. Here, we have added the corresponding text citation for this contribution as:

"[...], as the region contributes with around 15% of annual global fire carbon emissions (van der Werf et al., 2010)."

- ✓ Line 311: Should be *Tg yr$^{-1}$*.

We have corrected this throughout the manuscript.

- ✓ Figure 2: The colour bar used for the annual mean emission plots is not perceptually uniform and features large non-uniform breaks at arbitrary intervals. Please use an

appropriate colour bar – if these plots were made using Matplotlib, which they appear to have been, there are several available (see https://matplotlib.org/stable/users/explain/colors/colormaps.html). In addition, for the CO total column plots, the colour bar used is diverging which is inappropriate for displaying a continuous variable which is not a difference.

Thank you for bringing this to our attention. We aim to ensure clarity for a diverse range of readers, so we have updated the colour bars as suggested.

- ✓ Line 364: 'Andes' rather than 'Andean (mountain range)' is sufficient.
- ✓ Line 365: 'Accumulates'.
- ✓ Line 395: 'Despite the peak of precipitation being…', also there appears to be an incomplete citation here '(Grimm)'.
- ✓ Table 3: The caption should explain why some of the numbers are in bold type. There is also a more general question here about whether short-term trends of <10 years are meaningful given interannual variability and the complex political environment in South America (which the authors describe well in this section from a fire perspective).

We concur with Reviewers #1 and #2 that the 8-year period presents significant uncertainty due to the limited time sample and the high variability in emissions. This uncertainty is exacerbated by our use of annual CO emissions to calculate trends. Consequently, we have decided to exclude the short-term trend from our study.

Regarding Table 3, we added information in the footnote of the table describing the meaning of the trends in bold:

Note: The magnitudes of trends are highlighted in bold when the Mann-Kendall test indicates a significant trend at the 95% confidence level (p-value < 0.05). [...].

- ✓ Figure 6: This figure would benefit from hatching/stippling to denote where the biases are statistically significant according to an appropriate test.

Thanks for the suggestion, we included the hatching not only in that figure but also in other places where we wanted to highlight the significance of the calculated test. We decided using the hatching where the test was not significant, since this can shade the valuable information that significant values present.

- ✓ Line 509: 'Differed'.
- ✓ Line 515: 'Has', also 'simulated' rather than 'estimated'.
- ✓ Line 523: 'Increased' or 'enhanced', not 'extenuated' (this does not mean 'extended').
- ✓ Figure 8: Should be 'Spatio-temporal'.

This figure was changed to a simpler, more effective figure that shows the experiments' changes relative to the control run.

Lines 591-592: There are words missing and therefore the sentence does not read correctly. '…landscape fragmentation, which represents both….and can lead to…fire suppression effects'.

This sentence belongs now to the discussion session, as it was separated from the results section following Reviewer #2.1's comment. This sentence has been reworded as:

"In a fire model, crop representation needs to include the agriculture management cycle (Li et al., 2013), the influence of socioeconomic factors on management practices (Li et al., 2013),

agricultural expansion, and landscape fragmentation (Silva-Junior et al., 2022), among others. Crops representation would also need to include the agricultural role in fire suppression (Haas et al., 2022)."

- ✓ Line 602: Remove 'feature'.
- ✓ Line 619: This sentence is incomplete: 'Furthermore, agricultural expansion and landscape fragmentation.'. What about them?
- ✓ Line 645: 'TCCO' (although I suggest removing this variable entirely).
- ✓ Line 658: This should be '(broadleaf deciduous trees (BDT) and broadleaf evergreen tropical trees (BET-Tr))'.
- ✓ Line 662: 'Sensitivity'.
- ✓ Line 663: 'Small' rather than 'short'.
- ✓ Line 664: 'PFT'.
- ✓ Line 664-665: Awkward phrasing. Try 'Both improving PFT accuracy and incorporating representation of human land-use management, through variables such as land fragmentation, might help reduce biases'.
- ✓ Line 677: 'Cut'?
- ✓ Line 677: It should be noted that this code is only accessible to people with a Met Office account; for me it returned a login page. If the underlying model code is not publicly accessible, a statement is required to explain why; additionally, there is no link provided to the model output, which the journal also requires (or, in the absence of this, a statement explaining why the data are not being made publicly available).

We thank Reviewer #2 for pointing out the missing information required to access the model code. We also wanted to add that it is in our plan to share the model results via an open access platform such as Zenodo if this manuscript is accepted. To provide clear, detailed information for accessing the code, we have added the following in "Code and data availability" l 677:

The JULES-ES control configuration (based on JULES version 7.5) is stored at [https://code.metoffice.gov.uk/trac/roses-u/browser/d/l/3/2/3/trunk, last access:11 March 2025]. JULES and associated configurations are freely available for non-commercial research use, as set out in the JULES user terms and conditions [http://jules-lsm.github.io/access_req/JULES_Licence. pdf, last access: 10 November 2025]. For a comprehensive guide to accessing, installing, and running the configurations, we direct the reader to Appendix A in Wiltshire et al. (2020). Note that to view and use the JULES-ES source code, access will be required via the Met Office Science Repository Service [https://code.metoffice.gov.uk/trac/home, last access: 10 November 2025], and is available to those who have signed the JULES user agreement. The easiest way to access the repository is to complete the online form to register at [http://jules-lsm.github.io/access_req/JULES_access.html, last access: 10 November 2025].

- ✓ Line 678: 'Are downloaded'; also, I suggest putting the dataset links in brackets.
- ✓ Line 686: This doesn't make sense: 'Some assessments were done using the deforestation front for 2020 provided at and the ecoregion provided at'.
- ✓ There are some typos in the reference list, and the DOIs are inconsistently stated with some references missing DOIs (e.g. Magahey and Kooperman 2023) and others having double entries e.g. line 872: https://doi.org/https://doi.org/10.1007/s10531-019-01720-z. Presumably these are artefacts introduced by reference management software but the reference lists that these produce should always be checked manually.

We thank Reviewer #2 for their detailed review. We have checked the reference list to avoid duplicate DOIs or missing information.

**Added References**

Lichtig, P., Gaubert, B., Emmons, L. K., Jo, D. S., Callaghan, P., Ibarra-Espinosa, S., Dawidowski, L., Brasseur, G. P., and Pfister, G.: Multiscale CO Budget Estimates Across South America: Quantifying Local Sources and Long Range Transport, Journal of Geophysical Research: Atmospheres, 129, e2023JD040 434, https://doi.org/10.1029/2023JD040434, 2024.

Power, J., Côté, M.-P., and Duchesne, T.: A Flexible Hierarchical Insurance Claims Model with Gradient Boosting and Copulas, North American Actuarial Journal, 28, 772–800, https://doi.org/10.1080/10920277.2023.2279782, 2024.

Reddington, C. L., Morgan, W. T., Darbyshire, E., Brito, J., Coe, H., Artaxo, P., Scott, C. E., Marsham, J., and Spracklen, D. V.: Biomass burning aerosol over the Amazon: analysis of aircraft, surface and satellite observations using a global aerosol model, Atmospheric.

Chemistry and Physics, 19, 9125–9152, https://doi.org/10.5194/acp-19-9125-2019, 2019

Wiltshire, A. J., Duran Rojas, M. C., Edwards, J. M., Gedney, N., Harper, A. B., Hartley, A. J., Hendry, M. A., Robertson, E., and Smout-Day, K.: JULES-GL7: the Global Land configuration of the Joint UK Land Environment Simulator version 7.0 and 7.2, Geoscientific Model Development, 13, 483–505, https://doi.org/10.5194/gmd-13-483-2020, 2020.